# Large Language Models' Expert-level Global History Knowledge Benchmark (HiST-LLM)

**Jakob Hauser**[1,†]**, Daniel Kondor**[1]**, Jenny Reddish**[1]**, Majid Benam**[1]**, Enrico Cioni**[2]**, Federica Villa**[2]**,**
**James S. Bennett**[3]**, Daniel Hoyer**[4]**, Pieter Francois**[2,5]**, Peter Turchin**[1]**, R. Maria del Rio-Chanona**[1,6,†]

[1] Complexity Science Hub, Vienna, Austria    [2] University of Oxford, Oxford, UK
[3] University of Washington, Seattle, WA, USA    [4] George Brown College, Toronto, Canada
[5] The Alan Turing Institute, London, UK
[6] Department of Computer Science, University College London, London, UK
[†] hauser@csh.ac.at; m.delriochanona@ucl.ac.uk

## Abstract

Large Language Models (LLMs) have the potential to transform humanities and so-cial science research, yet their history knowledge and comprehension at a graduate level remains untested. Benchmarking LLMs in history is particularly challeng-ing, given that human knowledge of history is inherently unbalanced, with more information available on Western history and recent periods. We introduce the History Seshat Test for LLMs (HiST-LLM), based on a subset of the Seshat Global History Databank, which provides a structured representation of human historical knowledge, containing 36,000 data points across 600 historical societies and over 2,700 scholarly references. This dataset covers every major world region from the Neolithic period to the Industrial Revolution and includes information reviewed and assembled by history experts and graduate research assistants. Using this dataset, we benchmark a total of seven models from the Gemini, OpenAI, and Llama families. We find that, in a four-choice format, LLMs have a balanced accu-racy ranging from 33.6% (Llama-3.1-8B) to 46% (GPT-4-Turbo), outperforming random guessing (25%) but falling short of expert comprehension. LLMs perform better on earlier historical periods. Regionally, performance is more even but still better for the Americas and lowest in Oceania and Sub-Saharan Africa for the more advanced models. Our benchmark shows that while LLMs possess some expert-level historical knowledge, there is considerable room for improvement.

## 1   Introduction

Large Language Models (LLMs) have the potential to simulate human behavior [1] [2], label data, perform legal tasks [3], extract information on historical events [4], perform thematic analysis [5], advance psychological measurement [6], among other applications [7]; thus many authors expect LLMs to revolutionize social science and humanities research [8]. However, to fully reap the benefits of LLMs we must also overcome biases and unbalanced distribution of knowledge that may originate from inherent biases in training datasets [9]. In this work, we introduce a unique data set of graduate and expert academic level global history knowledge, Seshat: Global History Databank [10–12]. We make this dataset publicly available and use it to benchmark historical knowledge of large language models. We name this benchmark History Seshat Test for LLMS (HiST-LLM)[1].

Our knowledge of history is inherently unbalanced and partial, complicating the fair measurement of LLMs' historical knowledge. We know more about the history of the Global North than the Global

---

[1]Results and dataset can be found at `https://github.com/seshat-db/HiST-LLM` and `https://doi.org/10.5281/zenodo.14671247`

South, and recent periods are better documented than ancient ones. This imbalance is likely reflected in LLMs; for instance, GPT-3.5 and GPT-4 perform excellently on US history at a high school level but only reasonably well on global history [13][2]. There are also several gaps in our knowledge about the past. Consequently, experts and historians make inferences based on context and other known information. This in turn can lead to disagreements among even professional historians; for instance, estimates of the peak population of the Roman Empire range from 50 to over 150 million [14]. At a graduate and expert level, we may expect the knowledge gap between Global North and Global South countries to increase given that the former has been researched more widely than the latter [15, 16]. This imbalance makes it difficult to assess how well LLMs represent historical knowledge. How can we measure their balance of knowledge when our own knowledge is unbalanced?

To perform a fair assessment of LLMs' historical knowledge, we need a systematic compilation of existing historical knowledge across regions. However, compiling such data poses several challenges. First, given the wide range of theoretical questions and approaches in history and archaeology, deciding which variables to record and whether they are best conceptualized as attributes [17] or events [18] and whether they should be recorded as numerical or categorical already poses a challenge. Second, covering more than one or a few regions across different time periods is challenging since experts in history and archaeology specialize on particular regions and time periods. Therefore a comprehensive regional and temporal coverage requires engaging with multiple experts and academic resources. Third, most history and archaeology research focuses on recording well-established facts, but scarcely records that a vast amount of historical knowledge is inferred.

To address these challenges, we present the Seshat Global History Databank: a structured and systematic dataset that aims to capture the state of historical knowledge worldwide. The Seshat Databank has a hierarchically structured codebook with numerical and categorical values, recorded for over 600 unique past societies ("polities") spread across all UN geographic regions [3]. All variables are recorded as "evidenced" or "inferred", with data collected over more than a decade in collaboration with expert historians and archaeologists. Each variable's value is linked to at least one academic or expert reference and a textual explanation. Overall, the dataset used in the current article comprises 383 variables with 36,000 data points across more than 600 polities, and over 2,700 references.

We use the Seshat Databank to go beyond questions that are part of "general knowledge" and focus on history on the graduate and academic level. To test LLMs' knowledge, we convert the dataset into multiple-choice questions that we then feed into LLMs via prompts. An important component of our benchmarks is that we not only test whether LLMs can identify correct facts, but our questions also explicitly ask whether a fact is evidenced or inferred based on indirect evidence. We find that while LLMs outperform random guessing, they fall short of expert comprehension. LLMs' knowledge is relatively evenly spread between global regions. Nonetheless, OpenAI's models perform better for Latin America and the Caribbean, while Llama performs best for Northern America. Both OpenAI's and Llama models' performance is worse for Sub-Saharan Africa. Across time, there is a clear pattern: LLMs perform best for earlier periods.

## 2 Related Work

### 2.1 History and anthropology datasets

This work is part of and builds upon the Seshat Global History Databank project, which has collected historical data since 2011 with the purpose of documenting what is currently known and unknown about the social and political organization of past human societies across the globe and how they have evolved over time [11, 12]. To date, these data have been used primarily to test different historical theories and hypotheses about long-term cultural evolution and the role of agriculture, warfare and religion in human history [19–23]. While Seshat kept records of references and written justifications, previous permanent [4] releases of these data did not link each data point to an academic or expert source in a machine-readable way nor make the explanatory text readily usable for machine processing, and were not used to benchmark or train LLMs.

---

[2]GPT-4 scored above 80% in Advanced Placement US History and Advanced Placement Art History, but only above 60% in World History.

[3]`https://unstats.un.org/sdgs/indicators/regional-groups/`

[4]Alongside [10] a public wiki-type page was released with some text descriptions. However this was then replaced by permanent Zenodo releases with only variable values.

In addition to Seshat, there have been other global data collection efforts for anthropology. Notable examples are the Standard Cross-Cultural Sample and the Ethnographic Atlas [24–27]. Originally published as printed volumes, and later as digital data tables, these datasets were created for comparative research among societies with diverse environmental and social conditions, and thus include a wide range of categorical variables whose values were assigned by experts, based on a reading of a wide literature that typically included textual descriptions of the societies in question. Recently, D-PLACE was created as a digital, online resource that collects these and related datasets [28] and allows quantitative and statistical analysis of a wide range of questions. In total, D-PLACE contains over 600 thousand data points, each of which represents an individual "fact" about a certain society. However, these datasets mostly focus on small-scale societies (tribes and chiefdoms between 100 and 100,000 people), do not link specific facts to their sources individually, do not have textual explanations, and do not record changes of variables across time, nor when a variable is unknown or disputed.

Our work also relates to datasets that focus on specific historical topics. The Database of Religious History (DRH) collects knowledge about historical and modern religious practices based on expert knowledge [29–31]. The latest version contains over 250,000 data points, each pertaining to one specific aspect of religious practice of a specific group or setting. Data points are supported by either expert comments (textual descriptions) or citations to academic literature. Another specific topic dataset is PAGES, which compiles data on past land use, a valuable input for models that connect human activities to environmental conditions [32]. Focusing on modern times, the Correlates of War project gathered detailed data about military conflicts from the past two centuries, including several categorical variables and numerical estimates for each of hundreds of conflicts included [33].

Last, our work relates to datasets that do not record events, but compile original historical texts. These include the Perseus Digital Library, a corpus focusing mainly on classical (Mediterranean) antiquity, with a total corpus size of over 68 million words, along with English translations for a large part of the corpus [34, 35]; the Trismegistos project that provides a database of all known names from classical (Mediterranean) antiquity and references to their sources [36]; the Papyrological Navigator, a digitized corpus of texts written on papyrus, typically from the dry areas of the Mediterranean [37]; or the ORACC corpus of cuneiform texts [38]. While traditionally these have been the domain of specialists who would work on reconstructing, translating and interpreting them, recent advances in natural language processing (NLP) techniques provide several promising applications [39, 40]. Languages with larger corpora, such as Latin and Akkadian, have already been used for training language models with a realistic aim of machine-assisted translations [41, 42]; in the case of smaller corpora, machine learning models can still assist in more specialized tasks, such as the reconstruction of text fragments on inscriptions or papyri [43, 44]. These works could potentially be used as training or benchmark resources for LLMs, but currently lack the structure and broad focus to applications beyond linguistics or highly specialized applications.

## 2.2 Benchmarking LLMs knowledge

Our work is also related to the growing research benchmarking LLMs' knowledge and performance in specific disciplines. The MMLU benchmark of Hendrycks et al. [45] measured the performance of GPT-3 and related models across a wide range of disciplines using multiple-choice questions. Models generally performed better for social sciences and humanities than natural sciences. The benchmarks included separate categories for high-school level knowledge of European, US and world history; however, performance was similar across these categories, with accuracy scores between 50% and 65% for the best performing GPT-3 and UnifiedQA models[5]. Recent models have substantially improved upon this: GPT-3.5-Turbo achieved 85.7% accuracy, GPT-4 Turbo, the best current model, achieved 95.8% accuracy, and Llama-3-70b, the best open-weight model, achieved 94.1% accuracy on the High School World History subtask of MMLU [46]. However, because world history taught in high school is biased towards Western history [47], these results are not informative regarding the balance of LLMs knowledge in comparison to what is actually known of societies by expert historians. In contrast, our work compares the knowledge of LLMs across all expert knowledge.

Recent efforts have focused on benchmarking LLM performance in specific fields. For instance, Guo et al. [48] demonstrated LLMs' capabilities in various chemistry tasks. Guha et al. [3] confirmed their proficiency in legal tasks, and Gandhi et al. [49] evaluated their social reasoning abilities. Chen

---

[5]Each question had four choices, so a random performance in this case would translate to an accuracy of 25%

et al. [50] introduced a QA benchmark dataset for time-sensitive questions, relevant to historical knowledge. Several papers also address integrating general and domain-specific knowledge. Onoe et al. [51] presented the CREAK dataset, combining specific facts with common-sense reasoning, and a training procedure to enhance LLM performance. Ge et al. [52] offered a framework for interfacing general-purpose LLMs with domain expertise. These studies often use few-shot learning, leveraging models' ability to generalize from limited examples [51, 53].

Literature on biases in language models and AI systems is growing, assessing the values and norms they represent. Early research highlighted how biases in training data manifest in word embeddings and models [54, 55], with warnings about the potential for harms and discriminatory outcomes in algorithmic systems [56]. Recent studies propose complementary approaches to mitigate biases during LLM training and deployment, including explicit safety pipelines [13, 57], detailed data collection on social biases [58], and model benchmarks on their prevalence [59]. However, most focus on stereotypes and associations about specific groups rather than biases in knowledge distribution across space and time.

## 3 Dataset

The Seshat database contains historical knowledge dating from the mid-Holocene (around 10,000 years before present) up to contemporary societies. However, the bulk of the data pertains to agrarian societies in the period between the Neolithic and Industrial Revolutions, roughly 4000 BCE to 1850 CE. The information contained in the dataset runs the gamut from the most basic historical information (e.g., the presence or absence of writing) to highly complex topics (e.g., certain religious and ideological systems) for which it is vital to record differing interpretations, provide nuance, and historical context. Seshat encodes what is known about global history, but also what is unknown, poorly known, and contested [10–12].

The dataset covers all UN geographic regions, with data collected about polities that occupied certain "Natural Geographic Areas" (NGAs) in them over time. In total, the Seshat team defined 35 NGAs for the purpose of data collection. For an illustration of the the distribution of data points across time, geographic regions and NGAs, see Figure S2 in the Appendix. In the current paper, for the purpose of studying performance across geography, we use a regional division scheme based on the UN regions, with two important modifications: (i) We split North America and Europe into two regions; (ii) We include Hawaii in Oceania. We made these modifications since most of the focus of Seshat is on polities that existed before colonization and Europe and North America had very different historical pathways in that period. Similarly, the first inhabitants of Hawaii are of Polynesian descent and thus until the nineteenth century, its history was more related to other regions in Oceania. We present the location of NGAs, along with the regional division scheme in Fig. 1. Overall, the dataset used in the current article comprises 383 variables and 36,000 data points, and over 2,700 references.

### 3.1 Data Format

For each polity, data is coded for a set of variables clustered around 11 themes or research topics, which we denote *Variable category*. For instance, there are variables describing the social complexity of polities – levels of administrative hierarchy, territorial scale of polities, information technology – as well as variables for military technology and organization, religion and rituals, agricultural techniques and productivity, institutions such as court systems and measures of biological well-being such as life expectancy or average adult stature. While the target of coding is individual polities, changes in variable values within a polity's lifespan are also recorded and time-stamped; for instance, noting a polity's population as 12 million from 100 to 150 CE, then 14 million from 150 to 220 CE.

Each datapoint ("Seshat record") consists of a coded value for a certain variable, linked to a named polity and date range, a justification text, which describes the evidential basis for the code, and citations to reliable scholarly source(s) including personal communication from an expert. Where no date range is given, the full date range of the polity is implied. Sometimes the justification texts include direct quotes of copyrighted material, in this cases, we keep the first 5 and last 5 words as indicators and replace the middle text with ellipsis "<...>". In the subset of the Seshat databank we use for Benchmarking possible values of data points are:

- Absent, present: evidence points to the absence or presence of a certain variable

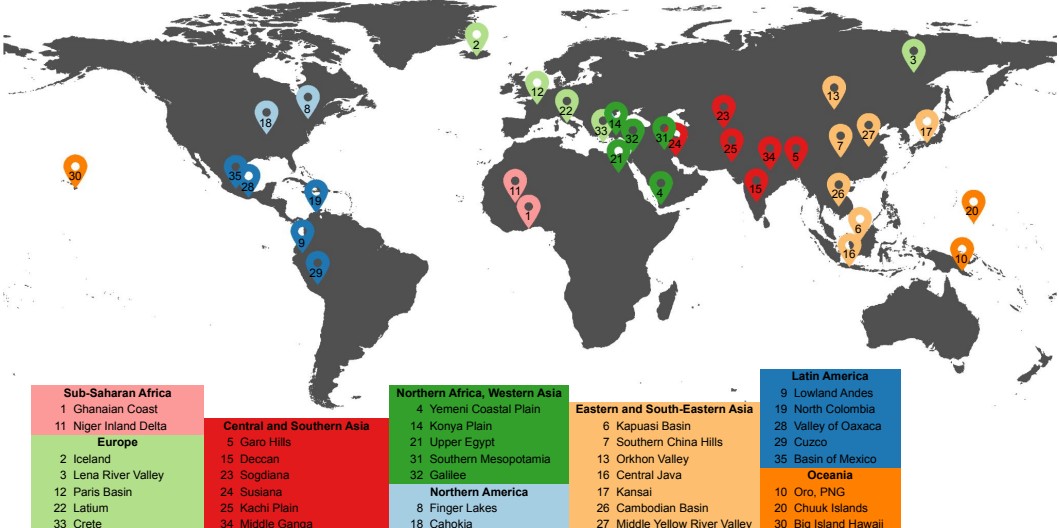

Figure 1: World map displaying our division of regions inspired by the UN geographic regions. Each marker on the map corresponds to an NGA, as defined by Seshat experts for the purpose of data collection. Researchers identified and collected data about each polity that occupied or overlapped with each NGA over the course of history. Colors correspond to the regional division scheme used in the current paper based on the UN geographic regions.

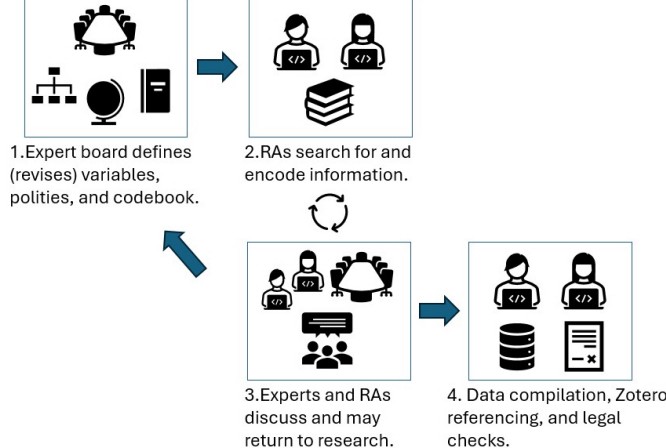

Figure 2: Illustration of Seshat's data collection process. RA stands for research assistants. Circular arrows denote iterative process and discussion. The blue arrow from step 3 to step 1 indicates that sometimes more polities and variables were added to the dataset and collection process after discussions among the Seshat team as well as external academic experts.

- Inferred absent, inferred present: Direct evidence is lacking or sparse, but indirect evidence points towards an absent or present code.

## 3.2 Data Collection

The data collection process comprises four main steps, illustrated in Figure 2. First, the Seshat Board[6] defines the variables and polities for study. The board aims to ensure spatial balance in the polities included in the sample, representing societies from all inhabited continents and UN geographic regions. To select variables, the board focuses on outstanding and contested research questions in

---

[6]https://seshatdatabank.info/seshat-about-us/seshat-who-we-are

history and archaeology, asking which data is needed to answer them. The resulting variables are compiled into a Codebook.

Second, data collection begins. Under the supervision of established scholars (professors and PhD-level researchers), research assistants (RAs) populate the database with accessible information from scholarly publications. While coding variables, RAs write narrative paragraphs explaining the rationale for each coding decision, citing relevant scholarship. While the language of these scholarly publications varies, there is a predominance of English sources. This mostly is due to the inherent imbalance in the language of history scholarship in the Global North, which is increasingly in English. The Seshat team is currently working on expanding the number of non-English sources.

Third, for variables used in previous Seshat publications [20, 22, 23], experts review and provide feedback. This feedback comes through various methods, including reviewing coded data sheets, emails, video meetings, and in-person workshops. Steps 2 and 3 are iterative, often requiring multiple rounds of expert feedback and revisions. Variables in the Codebook are also revised during these discussions, as indicated by the arrow from step 3 to step 1 in Figure 2. In the current dataset, 14,000 out of 36,000 data points underwent this process of expert review and cleaning prior to being made publicly available online [60] and used in articles published by the Seshat team. The remaining 22,000 were coded by graduate-level research assistants under the supervision of the Seshat board but have not yet been subjected to the same level of scrutiny. However, the selection of variables for use in articles was guided by research questions, which have focused on social complexity and warfare, rather than quality of data, and past experience indicates that only a small proportion of data (less than 5 percent) will change as a result of expert review. The review status of each data point is indicated in a dedicated column of the released dataset.

In the fourth step, we assemble and curate all data, ensuring each reference is uniquely linked to a Zotero database. For the current publication, we have avoided copyright infringement by making publicly available only narrative paragraphs that do not directly quote from references.

## 3.3   Data Limitations and Ethical Considerations

The Seshat database has several limitations. First, the data was collected mostly, though not exclusively, from English-language sources. This limitation likely results in less comprehensive coverage for non-English-speaking regions and limits diversity which is an ethical consideration and an area of improvement. Second, while the Seshat Board aimed for broad cross-cultural applicability when defining variables, their choices may reflect biases related to their expertise and backgrounds, and research assistants may have subjective interpretations when coding variables. Third, new evidence can update historical knowledge, so this benchmark reflects only the current understanding. However, the team consulted a wide range of experts for diverse perspectives and continuously revises the dataset. Regarding ethical considerations, users should be aware of copyright issues; while the dataset accompanying the current article is released under creative commons license, going beyond it, and scraping a considerable amount of text from cited references to train models should only be done in consultation with a legal team and ethical board approval to avoid infringing copyright of the cited references.

Seshat takes special care in the data collection process, providing detailed qualitative justifications and soliciting expert feedback; however, the dataset may still simplify important historical knowledge (see Slingerland et al. [61] for issues and solutions in creating systematic datasets for cultural comparison). Therefore, using this dataset for historical analysis and interpretation should only be done in collaboration with relevant experts [11, 61, 62]. While the Seshat dataset has been used in several quantitative Cliodynamic[7] studies [19, 22, 23], it remains an independent dataset that can be used for different research approaches and is not tied to any single theoretical framework.

We acknowledge that the Seshat project involves the study of myriad different communities and populations from the past. Some peoples living today trace their ancestry to one or more of these past groups. As researchers, we have an obligation to present fair-minded, responsible, and respectful information concerning the past. While maintaining a commitment to scientific enquiry, we are committed to avoiding biased interpretation or representation of past or contemporary cultures, to refraining from using harmful or disrespectful terminology, and to treating sensitive information

---

[7]Cliodynamics is an interdisciplinary field that combines historical analysis and mathematical modelling. For more details and criticism see [62–65]

or topics with appropriate nuance and respect for the dignity and lived experiences of descendant communities.

## 4  Evaluation Process Setting

We benchmark the historical knowledge of chat-based models GPT-3.5, GPT-4-Turbo, GPT-4o, Llama-3-70b, Llama-3.1-70B (FP8), Llama-3.1-8B (FP8) and Gemini-1.5-flash. We selected these models for their state-of-the-art capabilities, and accessibility. While our model selection is not exhaustive, it includes a balanced set of both open and closed weight models. All models are queried using temperature 0 and for OpenAI models we set the random seed to 42. Models are evaluated by checking if their answers match exactly with the correct answers. Our evaluations were executed in August 2024, and for OpenAI models, we report model fingerprints in the dataset.

To benchmark the LLMs' historical knowledge, we converted the dataset into multiple-choice questions and prompted LLMs to answer these. Specifically, we asked whether a certain variable (e.g., writing) was present, absent, inferred present, or inferred absent in a particular polity and time frame. We used a multi-shot approach, providing four examples demonstrating solutions to other questions, in order to aid the models understanding of the task and additionally ask the model to provide its own reasoning before giving an answer. The exact prompts alongside an extended description of the conversion process can be found in Appendix B. We also used personification, instructing the LLMs to act as history experts to enhance their performance. Although data contamination is a central issue for benchmarks, we believe it is less critical for our benchmark, which primarily assesses the ability of LLMs to retain factual knowledge. In our evaluation, we exclusively utilized APIs that prevent the leakage of our benchmark to cloud providers and AI companies.

Given the unbalanced distribution of answers in our dataset[8], we measure performance using balanced accuracy, which is standard in several benchmark studies [3, 48]. We also use adjusted balanced accuracy, normalizing the metric between 0 and 1, where 0 represents the average accuracy of random guessing. To assess whether differences in performance across models, time, regions, and categories were significant, we used bootstrap resampling (with 1000 resamples) to estimate the standard error of the balanced accuracy. We report 95% confidence intervals in all our results. We analyze the LLMs' overall performance across UN World regions, time periods, and Seshat variable categories. To determine whether the limitations of the models pertain to general knowledge or the distinction between inferred and known facts, we also measure balanced accuracy using only the "Present" and "Absent" categories. We refer to this as the 2-choice balanced accuracy, whereas the test including inferred categories is the 4-choice balanced accuracy.

## 5  Results

With respect to the entire dataset, all models exhibit similar performance, with balanced accuracy ranging from 37.3% for GPT-3.5 to 43.8% for GPT-4-Turbo (random performance would result in 25%). Table 1 shows the balanced accuracy and adjusted balance accuracy of all models across the whole dataset. This table also includes the benchmark in a 2-choice format (absent/present), for which models balanced accuracy ranges from 57.6 to 63.2 (random performance would result in 50%).

Overall, GPT-4-Turbo performs the best and Llama-3.1-8B the worst in the 4-choice test. Gains in performance are related to model size, with the GPT-4 based models having a 10%-15% advantage over the much smaller GPT-3.5-turbo and Llama-3.1-8B. The larger Llama models and Gemini are placed in between. In the 2-choice balanced accuracy, GPT-4o performs best with a score of 63.2%, while Llama-3.1-8B performs worst. Our results indicate that while all models perform significantly above random, they still fall short of encompassing expert-level knowledge of history.

### 5.1  Comparison across time, regions, and categories

Here we explore the performance of models across time, regions, and categories. In Tables 2-4 we display results for the better performing models: the two variants of GPT-4, Llama-3.1-70B (FP8)

---

[8]Our dataset is unbalanced across each category with 47% "Present", 20% "Absent", 16% "Inferred Present", and 17% "Inferred Absent".

Table 1: Overall performance of models with chain of thought prompting and category specific few-shot examples, measured by 2-choice and 4-choice balanced accuracy and adjusted balance accuracy. Brackets show confidence intervals using bootstrap resampling. Bold entries highlight the best performing model.

| Model | Balanced accuracy | | Adjusted balanced accuracy | |
|---|---|---|---|---|
| | (4-choice) | (2-choice) | (4-choice) | (2-choice) |
| Gemini-1.5-flash | 39.0 [38.5, 39.5] | 60.2 [59.7, 60.7] | 18.7 [18.0, 19.3] | 20.5 [19.4, 21.5] |
| GPT-3.5-turbo | 35.6 [35.2, 35.9] | 57.9 [57.4, 58.4] | 14.1 [13.5, 14.6] | 15.7 [14.7, 16.7] |
| GPT-4-turbo | **46.0** [45.5, 46.5] | 60.0 [59.5, 60.6] | **28.0** [27.3, 28.6] | 20.0 [19.0, 21.0] |
| GPT-4o | 44.7 [44.2, 45.2] | **63.2** [62.7, 63.6] | 26.3 [25.7, 26.9] | **26.4** [25.4, 27.3] |
| Llama-3-70B | 39.5 [39.2, 39.9] | 62.6 [62.1, 63.1] | 19.4 [18.8, 20.0] | 25.3 [24.3, 26.3] |
| Llama-3.1-70B (FP8) | 40.8 [40.4, 41.2] | 62.6 [62.1, 63.1] | 21.1 [20.5, 21.6] | 25.2 [24.1, 26.2] |
| Llama-3.1-8B (FP8) | 33.6 [33.2, 34.1] | 57.6 [57.1, 58.1] | 11.5 [10.9, 12.1] | 15.2 [14.1, 16.2] |

and Gemini. In the Appendix, Tables S2-S4, we additionally include results for the other models, along with average performance across all seven and the number of data points that fall under each case.

We find some heterogeneities in model performance across categories. Most notably, model performance declines as the time frame approaches the present day, especially for models with overall better performance, including GPT-4-turbo and Llama-3-70B. See Table 2 for balanced accuracy over temporal bins and Figure S4 in the Appendix for a more graphical representation. Our results show that early historical periods, particularly those before 3,000 BCE, exhibit higher balanced accuracy scores for most models compared to more recent times. [9]

Table 2: Balanced accuracy over varying temporal ranges for four models. Rows correspond to time periods of 2,000 years before 4,000 BCE and 500 years after 4,000 BCE. Brackets show confidence intervals using bootstrap resampling. Bold values indicate the best performing model in that time range (across all seven models; note that for the time ranges between 8,000 and 4,000 BCE, the best performing models are not among the four displayed here). See Table S2 in the Appendix for the other three models, mean balanced accuracy scores across models and the number of data points in each time interval.

| Time range | Gemini-1.5-flash | GPT-4-turbo | GPT-4o | Llama-3.1-70B |
|---|---|---|---|---|
| 10k – 8k BCE | 33.3 [33.3, 50.0] | **42.9** [33.3, 62.5] | 38.1 [33.3, 50.0] | 33.3 [33.3, 50.0] |
| 8k – 6k BCE | 46.5 [40.9, 53.0] | 55.3 [48.0, 62.5] | 40.7 [35.3, 47.3] | 47.8 [42.1, 53.7] |
| 6k – 4k BCE | 44.4 [40.7, 48.5] | 48.1 [44.3, 51.8] | 41.9 [38.6, 45.6] | 46.7 [43.4, 50.0] |
| 4k – 3.5k BCE | 29.2 [21.5, 37.7] | **47.2** [36.3, 58.5] | 46.5 [36.1, 56.8] | 41.1 [30.5, 51.1] |
| 3.5k – 3k BCE | 43.4 [39.2, 47.4] | **53.5** [49.0, 58.2] | 48.6 [44.8, 52.3] | 47.0 [43.0, 50.3] |
| 3k – 2.5k BCE | 39.4 [35.9, 43.2] | 42.1 [38.4, 46.1] | **46.3** [43.5, 49.4] | 44.6 [41.6, 47.7] |
| 2.5k – 2k BCE | 41.0 [37.7, 44.6] | **48.6** [45.2, 52.3] | 47.3 [44.3, 50.3] | 41.0 [38.2, 44.1] |
| 2k – 1.5k BCE | 39.7 [37.0, 42.1] | 47.0 [44.3, 49.7] | **48.9** [46.8, 51.2] | 41.1 [38.9, 43.3] |
| 1.5k – 1k BCE | 40.3 [38.1, 42.4] | **52.5** [50.0, 55.0] | 48.3 [46.3, 50.2] | 41.9 [39.9, 43.9] |
| 1k – 500 BCE | 37.7 [35.8, 39.5] | **45.6** [43.6, 47.6] | 44.2 [42.6, 45.8] | 42.1 [40.4, 43.8] |
| 500 BCE – 0 | 39.4 [38.0, 40.8] | 45.1 [43.5, 46.7] | **45.6** [44.2, 46.9] | 41.2 [39.9, 42.5] |
| 0 – 500 CE | 40.2 [38.7, 41.9] | 45.5 [43.9, 47.2] | **46.3** [44.9, 47.8] | 42.3 [40.9, 43.7] |
| 500 – 1k CE | 37.8 [36.7, 39.1] | 44.2 [42.9, 45.5] | **44.8** [43.7, 45.8] | 40.3 [39.2, 41.3] |
| 1k – 1.5k CE | 36.9 [35.8, 37.9] | **43.8** [42.5, 45.0] | 43.0 [41.9, 44.1] | 39.3 [38.4, 40.3] |
| 1.5k – 2k CE | 35.5 [34.5, 36.6] | 38.7 [37.4, 40.0] | **39.7** [38.6, 40.8] | 36.5 [35.5, 37.4] |
| Mean | 39.0 | **46.7** | 44.7 | 41.7 |
| Std. deviation | 4.2 | 4.3 | 3.3 | 3.7 |

For instance, GPT-4-Turbo achieves a balanced accuracy of 55.3% for the period 8,000 BCE – 6,000 BCE, but this drops to 38.7% for the period 1,500 CE – 2,000 CE. Similarly, Llama-3-70B's

---

[9]While GPT-4o and Llama-3.1-8B achieve their best performance between 4,000 BCE and 0 CE, their performance still declines for most recent periods.

Table 3: Balanced accuracy across geographic regions for four models. Brackets show confidence intervals using bootstrap resampling. Bold values indicate the best performing model (out of all seven) in that region. See Table S3 in the Appendix for the other three models, mean balanced accuracy values across models and the number of data points in each region; also see Tables S5 and S6 for finer grained results on the level of NGAs.

| Region | Gemini-1.5-flash | GPT-4-turbo | GPT-4o | Llama-3.1-70B |
|---|---|---|---|---|
| Central, S Asia | 40.6 [39.6, 41.6] | **46.7** [45.6, 47.7] | 45.3 [44.3, 46.1] | 41.5 [40.5, 42.4] |
| E, SE Asia | 38.9 [37.9, 39.9] | **44.7** [43.5, 45.7] | 44.1 [43.2, 45.1] | 39.0 [38.0, 39.9] |
| Europe | 37.9 [36.7, 39.2] | **44.4** [43.0, 46.0] | 43.4 [42.2, 44.9] | 41.0 [39.8, 42.3] |
| Latin America | 39.2 [37.2, 41.0] | 49.2 [47.0, 51.3] | **49.4** [47.2, 51.6] | 43.3 [41.3, 45.2] |
| N Africa, W Asia | 38.6 [37.6, 39.5] | **46.1** [45.0, 47.3] | 43.8 [42.9, 44.8] | 40.4 [39.6, 41.4] |
| N America | 40.9 [38.4, 43.3] | **46.3** [43.7, 48.8] | 45.2 [42.7, 47.6] | 43.4 [41.2, 45.8] |
| Oceania | 32.8 [30.3, 35.1] | 38.0 [35.1, 40.8] | **40.5** [37.7, 43.5] | 37.3 [35.0, 39.8] |
| Sub-Saharan Africa | 34.9 [32.7, 36.9] | **42.6** [40.2, 44.9] | 40.8 [38.5, 43.1] | 35.8 [33.9, 37.7] |
| Best performance | N America | Latin America | Latin America | North America |
| Worst performance | Oceania | Oceania | Oceania | Sub-Sah. Africa |
| Mean | 38.0 | **44.7** | 44.1 | 40.2 |
| Std. deviation | 2.6 | 3.1 | 2.6 | 2.5 |

performance decreases from 55.8% in the period 8,000 BCE – 6,000 BCE to 35% in the period 1,500 CE – 2,000 CE. The confidence intervals in brackets show this decrease is significant. These results suggest that while historical knowledge of early periods may be limited, LLMs perform better in these periods once this limitation is accounted for.

With respect to regions, GPT-4-Turbo is the leading model in six of the eight regions assessed (see Table 3 and S3), though some differences are within confidence intervals. GPT-4o has the second best average performance and also the best performance in Latin America and Oceania. Models typically perform best in the Americas, with the exepction of Llama-3.1-8B (East and Southeast Asia), while they perform the worst in Oceania and Sub-Saharan Africa. Somewhat surprisingly, Europe is not the region where models perform best. While regional performance differences are moderate, they may indicate a bias towards the American continent and a neglect of Oceania and Sub-Saharan Africa. The decrease in performance from GPT-4-Turbo and GPT-4o to GPT-3.5 in Sub-Saharan Africa underscores potential biases in model training and development.

Analysis across different categories also has some heterogeneity across models (see Table 4 and S4). GPT-4-Turbo achieves the highest scores in seven out of ten categories, including "Cults and Rituals" and "Legal System". GPT-4o leads in the remaining three categories, specifically "Economy", "Social Complexity", and "Warfare" variables. Gemini and Llama-3-70b follow with relatively good performance in the "Social Complexity", "Warfare" and "Well-being" categories in the case of both models, and especially bad performance for "Economy" in the case of Llama.

## 6    Discussion and Conclusion

In this paper presents a dataset derived from the Seshat Global History Databank to benchmark the historical knowledge of LLMs. This dataset comprises 36,000 data points of historical knowledge across over 600 polities, with references and explanations for each data point. As LLMs are increasingly used for information retrieval and search, monitoring LLMs knowledge is of crucial avoid replicate existing geospatial and historiographical biases [66, 67]. The Seshat dataset is structured a representation of historical knowledge and was designed to have regional and temporal representativeness, explicitly working to avoid Eurocentrism and other biases, making it a good candidate for a first benchmark of LLMs' global history knowledge.

Our benchmarking results reveal that, while the overall performance of the LLMs on expert historical knowledge is better than random guessing, it falls short of comprehensive expert-level knowledge. GPT-4-Turbo outperforms the other models in overall knowledge. We observe significant variation in model performance across different regions and time periods. Models perform best for earlier

Table 4: Balanced accuracy per variable category for four models. Brackets show confidence intervals using bootstrap resampling. Bold values indicate the best performing model (out of all seven) for that category. See Table S4 in the Appendix for the other three models, mean balanced accuracy values across models and the number of data points for each variable category.

| Variable category | Gemini-1.5-flash | GPT-4-turbo | GPT-4o | Llama-3.1-70B |
|---|---|---|---|---|
| Cults and Rituals | 37.7 [35.9, 39.7] | **45.8** [43.8, 47.7] | 43.4 [41.4, 45.4] | 39.5 [37.6, 41.3] |
| Economy | 34.5 [30.9, 38.5] | 39.4 [35.6, 43.3] | **44.0** [40.6, 47.7] | 36.7 [33.7, 39.5] |
| Discrimination | 38.2 [34.8, 41.6] | **42.0** [38.8, 45.4] | 39.2 [36.2, 42.3] | 36.6 [33.5, 39.7] |
| Institutions | 39.3 [37.6, 41.0] | **47.6** [45.3, 49.6] | 42.5 [40.7, 44.2] | 42.4 [40.7, 44.1] |
| Legal System | 34.6 [32.5, 36.6] | **50.1** [47.8, 52.7] | 45.5 [43.3, 47.7] | 38.3 [36.2, 40.2] |
| Religion and Normative Ideology | 33.6 [32.7, 34.5] | **42.8** [41.7, 44.0] | 39.8 [38.8, 40.7] | 38.1 [37.3, 38.9] |
| Social Complexity | 42.4 [41.2, 43.5] | **48.7** [47.4, 49.9] | 47.3 [46.2, 48.3] | 45.2 [44.2, 46.2] |
| Social Mobility | 34.3 [31.0, 37.6] | 38.6 [35.4, 41.9] | **43.9** [40.6, 47.5] | 36.4 [33.4, 39.4] |
| Warfare variables | 42.2 [41.2, 43.2] | 45.5 [44.4, 46.6] | **45.5** [44.7, 46.4] | 41.2 [40.4, 42.1] |
| Well-Being | 43.1 [41.0, 45.4] | **46.7** [44.5, 49.1] | 46.1 [44.2, 47.9] | 43.3 [41.5, 45.1] |
| Mean | 38.0 | **44.7** | 43.7 | 39.8 |
| Std. deviation | 3.5 | 3.7 | 2.5 | 3.0 |

historical periods, particularly those before 3000 BCE, with accuracy declining as the timeframe approaches the present day. This suggests that while LLMs handle limited early historical knowledge well, they struggle with the complexities of more recent data. Regionally, GPT-4-Turbo leads in six of the eight regions assessed, with Llama-3-70b leading in Northern America and GPT-3.5 in Sub-Saharan Africa. The consistent underperformance in Sub-Saharan Africa highlights potential biases in LLM training and development, indicating a need for more balanced training data.

Going forward this first benchmark should be taken as a lower bound on historical knowledge performance. The Seshat team continues to expand the data sources with explicit efforts to consult and cite non-English literature. Important future work includes increasing collaborations with universities in the Global South and indigenous groups, which will contribute to a more accurate understanding of global history [16, 47]. For regions and countries such as Latin America, China, Japan, Egypt, and the Middle East with a wider scholarship on languages other than English, we also expect the stringency of the benchmark of historical knowledge to increase as more data is gathered. Future work could explore the relationship between LLMs training data and their performance on specialized datasets like Seshat, to better understand the influence of training data on model accuracy. In addition, assessing global historical knowledge across diverse populations, with balanced representation of societies worldwide, would be a valuable study for the humanities and provide a meaningful human benchmark for LLMs to compare to.

While the underwhelming performance of the LLMs on expert historical knowledge may be disappointing, there is a silver lining. The dataset includes references and indicates the part of the reference where the information was gathered from. With proper legal and ethical checks, this dataset may be used to train models to compile future data. Given recent research showing declines in public data sharing after the release of ChatGPT [68] and the challenges of training LLMs with synthetic data [69], this public dataset could be an important step for improving LLMs history knowledge. The Seshat project continues to collect and review data, and it can be amplified by LLMs and expert review. This dataset can serve as a foundation for future applications, including training data acquisition models to automate the collection of historical data. By leveraging LLMs for preliminary data gathering and having experts review and refine the information, we can enhance both the efficiency and accuracy of historical data collection. Overall, while our results highlight areas where LLMs need improvement, they also underscore the potential for these models to aid in historical research.

## Acknowledgments and Disclosure of Funding

JH, MB, and MDRC acknowledges funding from CLARIA-AT, in addition MDRC acknowledges funding from James S. McDonnell Foundation. DK and PT are grateful for the financial support from the Austrian Research Promotion Agency (ESSENCSE-FFG873927). This work was also supported

by an AHRC-grant "Data/Culture. Building sustainable communities around Arts and Humanities datasets and software", Alan Turing Institute (PI Pieter Francois) and a UK Aid grant "Freedom of Religion or Belief Leadership Network" (University of Oxford, PI Pieter Francois). We thank Harvey Whitehouse for helpful comments and discussions as well as the Seshat board and volunteer for their input in the data collection efforts.

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

## A    Additional data description and results

Figure S2 shows the distribution of data points across space and time, in particular darker colors indicate a higher number of data points, while white areas signify the absence of data points, which is more common for earlier periods. The image shows that more data is available for recent periods. Notably, there is little data for Oceania, which is expected since it was the last continent to be populated (with the exception of Antarctica).

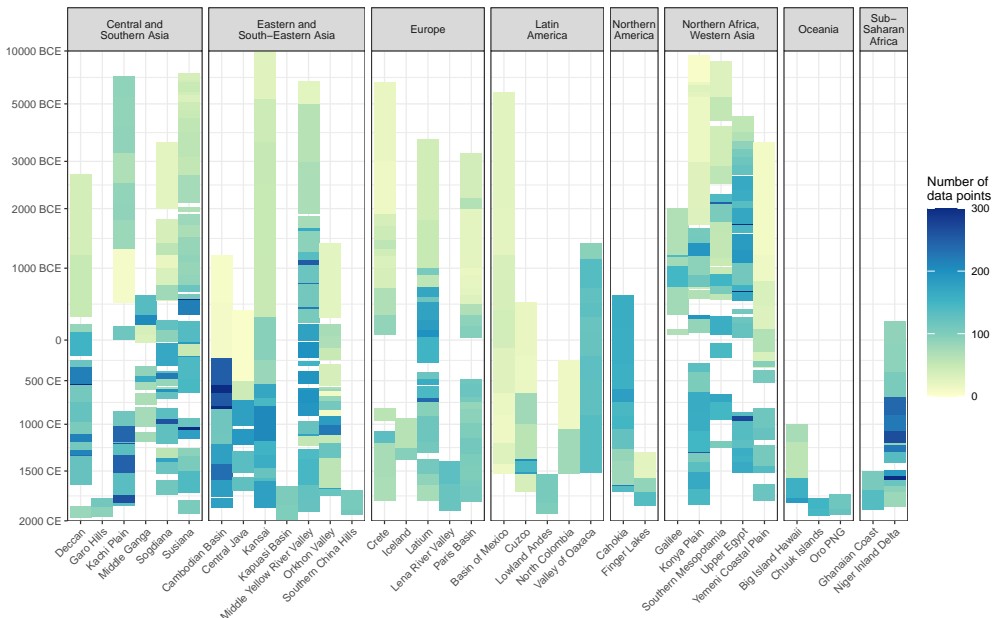

Figure S2: Heatmap indicates number of data points available for each NGA (bottom label) and geographic region (top label) over time (left axis).

Figure S3 shows the distribution of balanced accuracy for GPT-4-Turbo, the model with the best overall performance, across space and time. Darker colors indicate higher balanced accuracy, while completely white areas signify the absence of data points. Generally, more recent periods are colored more lightly, indicating lower accuracy of the models. Although one might assume that lower accuracy in more recent periods is due to more data being available, this is not necessarily true. For instance, in the NGA Basin of Mexico, there are roughly the same number of data points between 5000 BCE and 1000 BCE, yet the model's accuracy is higher for the earlier years.

To illustrate the result that earlier periods are, in general, better known by LLMs, we plot the balanced accuracy across time for the four models we analyzed in Figure S4. We also plot the number of data points available in black. As mentioned in the main text the bulk of the data pertains to agrarian societies in the period between the Neolithic and Industrial Revolutions, roughly 4000 BCE to 1850 CE. Our results show a decline in performance of LLMs across time. These results complement those in Table 2 in the main text. Additionally, in Table S2, we show mean balanced accuracy scores across all four models and the number of data points in each time period.

Figure S5 shows the distribution of answers obtained from the models and in the ground truth data. We see that models have significant differences among the answers given, most clearly in the case of GPT-4o which prefers the "inferred" both in the case of absence and presence of a variable.

In table 3 in the main text we showed the perfomance across UN regions. For completeness in table S5 we show the performance across NGA's.

Table S2: Results for three models over time, average accuracy across all seven models and the number of data points in each time range.

| Time range | GPT-3.5-turbo | Llama-3-70B | Llama-3.1-8B | Mean | Data points |
|---|---|---|---|---|---|
| 10k – 8k BCE | 33.3 [33.3, 50.0] | 42.9 [33.3, 60.0] | 29.2 [24.6, 44.4] | 36.1 | 34 |
| 8k – 6k BCE | 36.2 [31.0, 41.8] | **55.8 [50.8, 60.8]** | 33.1 [28.8, 38.3] | 45.1 | 349 |
| 6k – 4k BCE | 41.7 [38.4, 45.2] | **49.9 [46.9, 53.0]** | 35.2 [32.0, 38.7] | 44.0 | 613 |
| 4k – 3.5k BCE | 40.8 [31.1, 50.7] | 41.7 [31.8, 52.3] | 36.6 [27.2, 46.9] | 40.4 | 154 |
| 3.5k – 3k BCE | 35.9 [32.3, 39.5] | 47.6 [44.0, 51.4] | 39.1 [35.3, 42.9] | 45.0 | 464 |
| 3k – 2.5k BCE | 39.6 [36.6, 42.8] | 43.7 [40.7, 46.5] | 36.9 [33.9, 39.9] | 41.8 | 606 |
| 2.5k – 2k BCE | 36.8 [33.8, 39.7] | 41.1 [38.2, 43.8] | 36.7 [33.6, 39.8] | 41.8 | 807 |
| 2k – 1.5k BCE | 33.7 [31.6, 36.0] | 42.1 [39.9, 44.3] | 37.7 [35.4, 39.9] | 41.4 | 1308 |
| 1.5k – 1k BCE | 36.3 [34.4, 38.3] | 41.8 [40.1, 43.6] | 37.4 [35.4, 39.5] | 42.6 | 1830 |
| 1k – 500 BCE | 33.7 [32.0, 35.3] | 39.9 [38.3, 41.4] | 34.6 [32.9, 36.4] | 39.7 | 2499 |
| 500 BCE – 0 | 34.9 [33.7, 36.2] | 38.9 [37.6, 40.2] | 32.8 [31.4, 34.1] | 39.7 | 4029 |
| 0 – 500 CE | 35.2 [33.8, 36.5] | 40.0 [38.6, 41.4] | 33.4 [31.9, 34.9] | 40.4 | 3570 |
| 500 – 1k CE | 35.1 [34.1, 36.1] | 38.4 [37.5, 39.4] | 32.2 [31.1, 33.3] | 39.0 | 6367 |
| 1k – 1.5k CE | 35.7 [34.8, 36.7] | 38.1 [37.1, 39.0] | 32.8 [31.7, 33.9] | 38.5 | 7343 |
| 1.5k – 2k CE | 33.3 [32.4, 34.1] | 35.0 [34.1, 35.8] | 31.0 [30.0, 32.1] | 35.7 | 6604 |
| Mean | 36.2 | 42.5 | 34.6 | | |
| Std. deviation | 2.5 | 5.0 | 2.7 | | |

Table S3: Performance across world regions for three models, average accuracy across all models and the number of data points in each world region.

| Region | GPT-3.5-turbo | Llama-3-70B | Llama-3.1-8B | Mean | Data points |
|---|---|---|---|---|---|
| Central, S Asia | 36.6 [35.7, 37.6] | 40.6 [39.7, 41.4] | 33.5 [32.6, 34.5] | 40.7 | 8140 |
| E, SE Asia | 35.7 [34.8, 36.6] | 38.6 [37.7, 39.4] | 35.0 [34.1, 36.0] | 39.4 | 8432 |
| Europe | 33.9 [32.8, 34.9] | 39.3 [38.1, 40.6] | 32.5 [31.2, 33.8] | 38.9 | 4828 |
| Latin America | 37.8 [36.2, 39.6] | 42.1 [40.3, 44.0] | 33.8 [32.0, 35.5] | 42.1 | 2178 |
| N Africa, W Asia | 34.7 [33.7, 35.6] | 39.5 [38.6, 40.3] | 33.5 [32.6, 34.6] | 39.5 | 8525 |
| N America | 35.5 [33.6, 37.6] | 40.4 [38.2, 42.8] | 31.3 [29.2, 33.3] | 40.4 | 1915 |
| Oceania | 33.0 [30.8, 35.4] | 35.9 [34.0, 37.9] | 31.3 [29.0, 33.5] | 35.5 | 1012 |
| Sub-Saharan Africa | 37.3 [35.3, 39.1] | 35.1 [33.2, 37.1] | 29.2 [27.3, 31.3] | 36.5 | 1547 |
| Best performance | Latin America | Latin America | E, SE Asia | Latin America | |
| Worst performance | Oceania | Sub-Sah. Africa | Sub-Sah. Africa | Oceania | |
| Mean | 35.6 | 38.9 | 32.5 | | |
| Std. deviation | 1.6 | 2.2 | 1.7 | | |

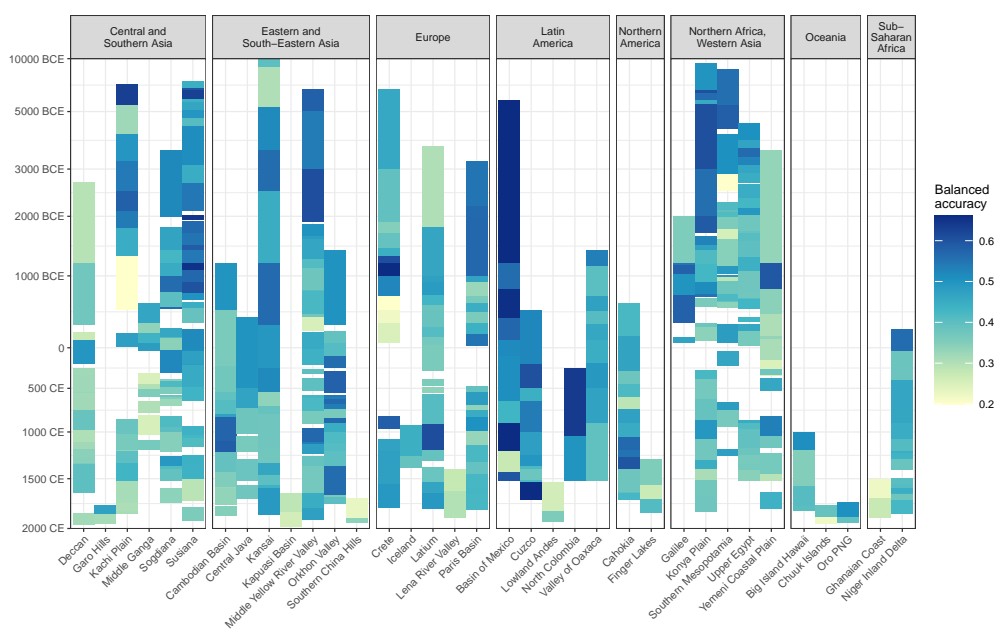

Figure S3: Heatmap indicates balanced accuracy score for each NGA (bottom label) and UN geographic region (top label) over time (left axis) for the GPT-4-Turbo model.

Table S4: Performance of three models, average accuracy across all models and the number of data points for each variable category.

| Variable category | GPT-3.5-turbo | Llama-3-70B | Llama-3.1-8B | Mean | Data points |
|---|---|---|---|---|---|
| Cults and Rituals | 35.0 [33.4, 36.6] | 36.8 [35.0, 38.8] | 32.8 [30.9, 34.6] | 38.7 | 3144 |
| Economy | 36.6 [32.8, 41.0] | 28.6 [26.3, 31.2] | 32.4 [28.6, 36.2] | 36.0 | 469 |
| Discrimination | 32.6 [29.8, 35.6] | 35.5 [32.3, 38.6] | 28.1 [25.4, 31.1] | 36.0 | 796 |
| Institutions | 36.6 [35.0, 38.3] | 41.6 [39.8, 43.4] | 36.7 [35.0, 38.4] | 41.0 | 2078 |
| Legal System | 31.4 [29.4, 33.5] | 33.9 [32.0, 35.7] | 29.2 [27.0, 31.6] | 37.6 | 1736 |
| Religion and Normative Ideology | 32.0 [31.3, 32.9] | 35.4 [34.7, 36.2] | 30.6 [29.7, 31.5] | 36.0 | 8334 |
| Social Complexity | 39.9 [38.9, 40.9] | 43.2 [42.3, 44.1] | 36.9 [35.8, 38.0] | 43.4 | 8167 |
| Social Mobility | 34.7 [31.6, 37.7] | 36.9 [33.9, 39.9] | 32.5 [29.5, 35.6] | 36.7 | 851 |
| Warfare variables | 38.1 [37.1, 39.1] | 42.0 [41.2, 42.8] | 34.0 [33.2, 34.9] | 41.2 | 9157 |
| Well-Being | 37.2 [35.1, 39.1] | 42.6 [40.7, 44.3] | 33.2 [31.2, 35.1] | 41.7 | 1845 |
| Mean | 35.4 | 37.7 | 32.7 | | |
| Std. deviation | 2.6 | 4.4 | 2.7 | | |

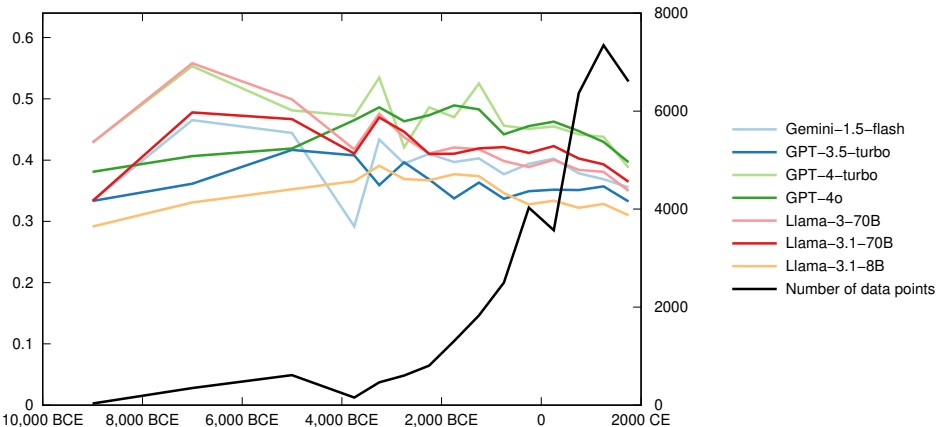

Figure S4: Spread of balanced accuracy score over time in colored lines. In black, the number of data points across time.

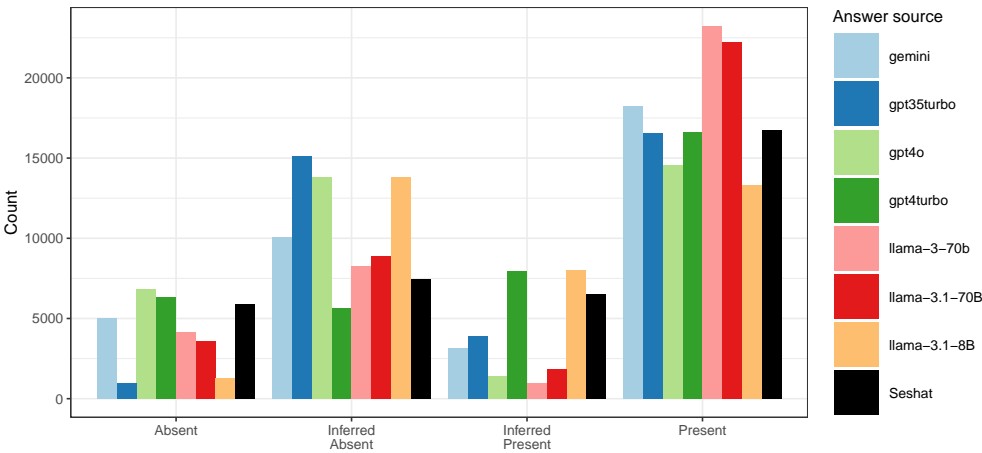

Figure S5: Counts of the Model wise predictions per label and of the labels in the Ground Truth

Table S5: Balanced accuracy per NGA and model

| NGA | Gemini-1.5-flash | GPT-4-turbo | GPT-4o | Llama-3.1-70B |
|---|---|---|---|---|
| Basin of Mexico | 45.7 [40.7, 51.1] | **53.1** [47.7, 59.2] | 50.9 [45.3, 56.6] | 44.4 [39.6, 49.3] |
| Big Island Hawaii | 37.7 [34.6, 40.9] | 41.9 [38.4, 45.9] | **45.1** [41.5, 48.4] | 43.3 [40.4, 45.9] |
| Cahokia | 40.9 [38.4, 43.6] | **45.9** [42.9, 49.0] | 45.2 [42.3, 48.1] | 43.7 [41.0, 46.2] |
| Cambodian Basin | 39.3 [37.0, 41.9] | **42.1** [39.6, 44.8] | 39.9 [37.6, 42.4] | 37.7 [35.4, 40.1] |
| Central Java | 34.3 [30.8, 37.8] | 40.1 [36.1, 44.4] | **42.9** [38.9, 46.8] | 41.7 [37.6, 45.6] |
| Chuuk Islands | 31.8 [25.8, 38.6] | 31.8 [26.3, 37.7] | **35.8** [29.6, 42.4] | 31.2 [27.2, 35.6] |
| Crete | 33.1 [29.7, 36.4] | **45.8** [41.9, 49.5] | 41.4 [38.1, 44.9] | 35.6 [32.5, 38.6] |
| Cuzco | 36.2 [32.9, 39.8] | 49.8 [44.8, 55.1] | **52.5** [48.3, 56.8] | 45.2 [41.4, 49.4] |
| Deccan | 38.8 [35.9, 41.6] | 37.0 [34.2, 39.7] | **45.4** [43.0, 47.9] | 40.1 [37.6, 42.5] |
| Finger Lakes | **41.2** [27.2, 55.0] | 39.5 [28.6, 53.9] | 30.3 [27.0, 34.3] | 34.3 [31.2, 38.0] |
| Galilee | 42.6 [37.0, 49.1] | **51.0** [43.6, 58.9] | 49.8 [45.0, 54.4] | 43.9 [39.0, 49.5] |
| Garo Hills | 29.3 [24.6, 42.6] | **45.5** [28.9, 61.8] | 31.6 [28.2, 44.8] | 43.1 [28.5, 58.2] |
| Ghanaian Coast | 35.3 [30.1, 41.1] | 25.2 [21.3, 29.9] | **39.2** [33.2, 45.9] | 34.4 [29.4, 40.4] |
| Iceland | 34.5 [28.4, 40.2] | **43.0** [34.6, 55.4] | 33.5 [29.2, 38.0] | 32.9 [28.2, 37.9] |
| Kachi Plain | 43.1 [40.5, 45.9] | **49.8** [47.2, 52.6] | 45.6 [43.2, 47.9] | 43.6 [41.5, 46.1] |
| Kansai | 38.7 [36.6, 40.8] | **45.6** [43.3, 48.2] | 45.0 [42.7, 47.2] | 38.4 [36.7, 40.2] |
| Kapuasi Basin | 28.2 [24.5, 32.3] | 29.0 [24.4, 35.5] | **32.3** [28.7, 38.2] | 29.2 [26.1, 33.2] |
| Konya Plain | 39.0 [37.0, 41.0] | **49.1** [46.6, 51.6] | 44.6 [42.6, 46.7] | 41.2 [39.4, 43.1] |
| Latium | 37.0 [35.0, 39.1] | **42.2** [39.8, 44.5] | 41.5 [39.6, 43.4] | 40.5 [38.8, 42.3] |
| Lena River Valley | 36.1 [30.2, 42.9] | 29.5 [22.5, 37.7] | **40.2** [33.3, 48.1] | 38.8 [33.7, 43.9] |
| Lowland Andes | 33.2 [28.6, 44.1] | 27.8 [22.9, 37.5] | **33.8** [30.3, 45.1] | 31.8 [28.5, 42.9] |
| Middle Ganga | 37.8 [34.6, 40.9] | 33.4 [30.7, 36.4] | **46.1** [43.1, 48.9] | 36.7 [34.3, 39.5] |
| Middle Yellow River Valley | 40.4 [38.6, 42.2] | 45.5 [43.5, 47.6] | **46.5** [44.9, 48.2] | 39.3 [37.7, 40.8] |
| Niger Inland Delta | 34.5 [32.1, 36.9] | **44.4** [41.9, 47.0] | 40.8 [38.3, 43.3] | 36.0 [33.8, 38.1] |
| North Colombia | 42.9 [35.0, 51.3] | **51.8** [41.5, 61.6] | 48.6 [40.1, 57.3] | 40.9 [32.9, 49.4] |
| Orkhon Valley | 38.8 [35.6, 41.8] | 43.3 [39.7, 47.0] | **44.6** [41.7, 47.4] | 36.5 [33.5, 39.4] |
| Oro PNG | 26.7 [22.5, 30.7] | **45.9** [34.9, 56.3] | 41.3 [31.2, 51.7] | 38.2 [30.0, 48.1] |
| Paris Basin | 41.4 [38.9, 43.9] | **48.5** [45.6, 51.4] | 47.7 [45.4, 50.2] | 44.3 [41.9, 47.0] |
| Sogdiana | 39.4 [36.9, 41.8] | **46.2** [43.1, 49.1] | 43.0 [40.7, 45.2] | 39.2 [37.1, 41.5] |
| Southern China Hills | **37.7** [30.5, 46.5] | 28.5 [21.6, 36.2] | 36.8 [31.1, 43.8] | 31.5 [27.2, 36.2] |
| Southern Mesopotamia | 38.8 [36.4, 41.2] | **46.0** [43.4, 48.8] | 41.8 [39.5, 44.1] | 38.3 [36.3, 40.5] |
| Susiana | 43.6 [41.8, 45.4] | **54.1** [52.3, 56.1] | 46.7 [45.2, 48.2] | 44.5 [42.9, 46.0] |
| Upper Egypt | 38.7 [37.1, 40.4] | **45.4** [43.7, 47.3] | 44.5 [43.0, 45.9] | 41.1 [39.6, 42.4] |
| Valley of Oaxaca | 38.5 [35.4, 41.8] | 46.8 [43.2, 50.2] | **48.1** [45.0, 51.1] | 44.7 [41.8, 47.9] |
| Yemeni Coastal Plain | 36.6 [33.5, 40.0] | **43.4** [39.5, 47.3] | 40.0 [37.3, 42.7] | 39.6 [36.9, 42.6] |
| Mean | 37.5 | **42.5** | 42.4 | 39.0 |
| Std. deviation | 4.3 | 7.6 | 5.5 | 4.4 |

Table S6: Balanced accuracy per NGA and model (continued), average accuracy and number of data points in each NGA.

| NGA | GPT-3.5-turbo | Llama-3-70B | Llama-3.1-8B | Mean | Data points |
|---|---|---|---|---|---|
| Basin of Mexico | 35.5 [32.1, 38.9] | 41.7 [36.7, 47.1] | 34.8 [31.0, 38.8] | 43.7 | 314 |
| Big Island Hawaii | 38.6 [35.3, 41.9] | 39.0 [36.6, 41.3] | 36.2 [33.1, 39.6] | 40.3 | 475 |
| Cahokia | 36.0 [33.7, 38.5] | 41.0 [38.3, 43.6] | 29.5 [27.0, 31.8] | 40.3 | 1666 |
| Cambodian Basin | 35.0 [32.9, 37.0] | 38.1 [35.9, 40.3] | 32.3 [30.1, 34.4] | 37.8 | 1781 |
| Central Java | 34.8 [31.4, 38.3] | 36.4 [32.7, 40.0] | 36.1 [32.3, 39.7] | 38.0 | 681 |
| Chuuk Islands | 26.7 [22.8, 30.6] | 34.3 [29.8, 39.9] | 26.2 [21.8, 30.6] | 31.1 | 298 |
| Crete | 35.6 [33.1, 38.1] | 35.2 [32.2, 38.8] | 27.8 [25.1, 31.0] | 36.4 | 844 |
| Cuzco | 37.6 [34.1, 41.1] | 44.5 [40.7, 48.3] | 30.0 [26.0, 34.1] | 42.3 | 434 |
| Deccan | 33.4 [31.0, 35.9] | 40.8 [38.2, 43.3] | 29.3 [26.7, 31.9] | 37.8 | 1300 |
| Finger Lakes | 32.0 [27.2, 38.6] | 31.8 [28.6, 37.9] | 29.6 [25.1, 36.4] | 34.1 | 249 |
| Galilee | 35.0 [30.2, 39.6] | 42.0 [37.3, 47.3] | 42.0 [35.3, 49.4] | 43.8 | 280 |
| Garo Hills | 38.3 [24.1, 52.7] | 36.2 [32.5, 50.3] | 24.7 [20.0, 35.4] | 35.5 | 204 |
| Ghanaian Coast | 32.9 [27.9, 38.6] | 32.9 [29.0, 37.4] | 32.1 [26.5, 38.5] | 33.1 | 243 |
| Iceland | 27.7 [22.9, 32.0] | 36.5 [32.2, 41.2] | 33.8 [27.9, 40.0] | 34.6 | 159 |
| Kachi Plain | 39.9 [37.9, 42.3] | 41.9 [39.8, 44.0] | 34.6 [32.5, 36.9] | 42.7 | 1471 |
| Kansai | 34.4 [32.7, 36.1] | 40.1 [38.3, 42.1] | 35.6 [33.6, 37.7] | 39.7 | 1703 |
| Kapuasi Basin | 31.2 [26.9, 37.5] | **32.5 [28.3, 37.6]** | 24.1 [19.9, 30.1] | 29.5 | 202 |
| Konya Plain | 36.1 [34.5, 38.0] | 39.3 [37.5, 41.1] | 33.3 [31.5, 35.3] | 40.4 | 2300 |
| Latium | 34.3 [32.7, 36.0] | 40.0 [38.4, 41.7] | 31.6 [29.5, 33.7] | 38.1 | 2003 |
| Lena River Valley | 29.2 [24.4, 34.8] | 31.2 [27.5, 35.0] | 29.7 [25.5, 33.8] | 33.5 | 252 |
| Lowland Andes | 29.3 [25.8, 39.6] | **37.5 [33.8, 51.1]** | 30.6 [24.9, 41.7] | 32.0 | 208 |
| Middle Ganga | 34.2 [31.3, 36.8] | 34.3 [31.8, 36.8] | 28.5 [25.6, 31.5] | 35.8 | 798 |
| Middle Yellow River Valley | 37.7 [36.2, 39.1] | 38.2 [36.9, 39.8] | 36.1 [34.2, 38.0] | 40.5 | 2815 |
| Niger Inland Delta | 38.5 [36.5, 40.6] | 35.0 [32.7, 37.2] | 28.5 [26.3, 30.7] | 36.8 | 1304 |
| North Colombia | 44.1 [35.5, 52.4] | 45.4 [38.4, 52.8] | 37.0 [29.4, 44.9] | 44.4 | 94 |
| Orkhon Valley | 31.4 [28.3, 34.2] | 35.1 [32.2, 38.0] | 32.1 [28.8, 35.5] | 37.4 | 1030 |
| Oro PNG | 28.5 [25.1, 31.7] | 36.1 [30.0, 43.9] | 27.2 [23.0, 31.8] | 34.8 | 239 |
| Paris Basin | 34.9 [33.1, 36.9] | 40.6 [38.3, 42.8] | 34.1 [31.8, 36.6] | 41.6 | 1570 |
| Sogdiana | 33.1 [31.1, 35.1] | 40.2 [37.8, 42.4] | 32.3 [29.9, 34.8] | 39.0 | 1521 |
| Southern China Hills | 26.8 [23.6, 30.2] | 31.5 [28.1, 34.9] | 29.1 [23.1, 35.3] | 31.7 | 220 |
| Southern Mesopotamia | 35.9 [33.7, 38.1] | 38.9 [36.5, 41.1] | 34.0 [31.7, 36.5] | 39.1 | 1624 |
| Susiana | 37.5 [36.0, 39.1] | 42.2 [40.7, 43.8] | 36.1 [34.4, 37.8] | 43.5 | 2846 |
| Upper Egypt | 33.1 [31.6, 34.5] | 40.2 [38.8, 41.6] | 34.5 [32.8, 36.2] | 39.6 | 3390 |
| Valley of Oaxaca | 37.9 [35.4, 40.5] | 41.4 [38.7, 44.2] | 35.2 [32.6, 38.3] | 41.8 | 1128 |
| Yemeni Coastal Plain | 33.8 [31.0, 36.7] | 38.5 [36.0, 41.1] | 29.2 [26.0, 32.7] | 37.3 | 931 |
| Mean | 34.3 | 38.0 | 31.9 | | |
| Std. deviation | 3.8 | 3.6 | 3.8 | | |

# B Question prompts

Here we describe the prompts used to query LLMs' knowledge of history. Prompt parts 1 and 2 provide the general instructions, emphasizing personification and instructing LLMs to respond with a short section of reasoning and one of the answers provided in the prompts. For each major question category we provide specifically tailored few-shot chain-of-thought examples. These tailored examples can be found in the prompt parts 5-14. As noted in the main text, the format for 'Cult and Rituals' variables differs slightly for grammatical accuracy. For these variables, we use prompt structured like part 4, and for all other we use 3. The main difference is the order in which the time frame is introduced and the inclusion of the phrase 'cults and rituals held by people'. The full prompt consists of prompt parts 1, 2, followed by one of 5-14 and a question structured in the same way as in 4 for 'Cults and Rituals', and 3 for all other variable categories.

---

**Prompt part 1** System Message

```
You are a history expert with extensive knowledge of global history.
```

---

**Prompt part 2** General Prompt

```
You will be tasked to answer multiple-choice questions concerning the historical presence or
absence of characteristics within the specified polity and time-frame, considering both
direct evidence and historical inference. Provide a short summary of the evidence, followed
by your reasoning, then answer the questions. The questions are always followed by the
corresponding answer options. Provide your answer by selecting the corresponding letter
from the options listed as follows: 'A: Present, B: Absent, C: Inferred Present, D: Inferred Absent'.
Keep your summaries as concise as possible. Even if you are unsure always provide an answer,
only provide the corresponding letter, and don't continue writing after giving an answer.
```

---

**Prompt part 3** Standard Benchmark Question

```
Question:
The characteristic 'Ideological thought distinguishes men and women' is categorized under
' Ideological reinforcement of equality '. Was it present, inferred present, inferred absent,
or absent for the polity called 'Late Cappadocia', during the time frame from 322 BCE to 93 BCE?
Options:
A: Present, B: Inferred Present, C: Inferred Absent, D: Absent
Reasoning and evidence:
```

**Prompt part 4** Cults and Rituals Benchmark Question

```
Question:
During the time frame from 1568 CE to 1603 CE, was the characteristic 'Mutilation', associated with the
cults and rituals held by people of the 'Japan - Azuchi-Momoyama' polity, present, inferred present,
inferred absent, or absent?
Options:
A: Present, B: Inferred Present, C: Inferred Absent, D: Absent
Reasoning and evidence:
```

**Prompt part 5** Few-Shot CoT Examples For Religion and Normative Ideology

```
Question:
The characteristic 'Ideological thought distinguishes men and women' is categorized under
' Ideological reinforcement of equality '. Was it present, inferred present, inferred absent,
or absent for the polity called 'Late Cappadocia', during the time frame from 322 BCE to 93 BCE?
Options:
A: Present, B: Inferred Present, C: Inferred Absent, D: Absent
Reasoning and evidence:

There is disagreement about the degree of sexual equality in Zoroastrianism.
According to more ancient Indo-Iranian religious traditions, women could not hope for
'ascent to paradise' but were consigned, along with slaves, to 'the kingdom of
shadows beneath the earth'.
Answer:
A

Question:
The characteristic 'supernatural enforcement of purity' is categorized under
'Normative Ideological precepts concerning morality/supernatural beings'.
Was it present, inferred present, inferred absent, or absent for the polity called
'Chuuk - Early Truk', during the time frame from 1775 CE to 1886 CE?
Options:
A: Present, B: Inferred Present, C: Inferred Absent, D: Absent
Reasoning and evidence:

Based on early 20th century accounts: Sharp Water Spirits could make people ill or "eaten"
or "bitten" by the sharp water if they did not follow certain rules, such as playing or
swimming too much in the same waters, not abstaining from sexual intercourse or not
abstaining from fish food before visiting the reefs where the spirits dwelled.
Answer:
B

Question:
The characteristic 'production of public goods' is categorized under
' Ideology reinforces prosociality '.
Was it present, inferred present, inferred absent, or absent for the polity called
'Roman Empire - Dominate', during the time frame from 285 CE to 394 CE?
Options:
A: Present, B: Inferred Present, C: Inferred Absent, D: Absent
Reasoning and evidence:

although great ideological force for prosociality and charity throughout this period, traditional
concern for 'civic duty' including public goods diminished as Christian ethic of
individual responsibility and prosociality (being moral, looking after less fortunate through alms)
become more prominent. Emperor and ruling class still engaged in public goods (urban infrastructure),
but less reinforced among non-ruling and provincial elites as it had been during principate
Answer:
C

Question:
The characteristic 'Automatic deification' is categorized under ' Rulers become gods after death '.
Was it present, inferred present, inferred absent, or absent for the polity called 'Achaemenid Empire',
during the time frame from 550 BCE to 331 BCE?
Options:
A: Present, B: Inferred Present, C: Inferred Absent, D: Absent
Reasoning and evidence:

There is 'no evidence' that Persian kings were deified after death.
Answer:
D
```

**Prompt part 6** Few-Shot CoT Examples For Social Complexity variables

```
Question:
The characteristic 'Knowledge/information buildings' is categorized under 'Specialized Buildings: polity
owned'. Was it present, inferred present, inferred absent, or absent for the polity called 'Fatimid
Caliphate', during the time frame from 909 CE to 1171 CE?
Options:
A: Present, B: Inferred Present, C: Inferred Absent, D: Absent
Reasoning and evidence:

Hall of Wisdom (Dal al-Hikma) built under al-Hakim in 1005 CE had "a very beautiful library." It was
accessible to all classes. Paper, pen, ink provided.
Answer:
A

Question:
The characteristic 'Professional military officers' is categorized under 'Professions'. Was it present,
inferred present, inferred absent, or absent for the polity called 'Proto-French Kingdom', during the
time frame from 987 CE to 1150 CE?
Options:
A: Present, B: Inferred Present, C: Inferred Absent, D: Absent
Reasoning and evidence:

during the reign of Philip I (1060-1108), the constable was one of the four "great officers" of the crown
Answer:
B

Question:
The characteristic 'Specialized government buildings' is categorized under 'Bureaucracy
characteristics'. Was it present, inferred present, inferred absent, or absent for the polity called
'Susiana - Early Ubaid', during the time frame from 5100 BCE to 4700 BCE?
Options:
A: Present, B: Inferred Present, C: Inferred Absent, D: Absent
Reasoning and evidence:

Wright and Johnson have argued that 'specialized governments' did not develop until the 4th millennium
BCE in southwestern Iran.
Answer:
C

Question:
The characteristic 'Fiction' is categorized under 'Kinds of Written Documents'. Was it present, inferred
present, inferred absent, or absent for the polity called 'Pre-Ceramic Period', during the time frame
from 7800 BCE to 7200 BCE?
Options:
A: Present, B: Inferred Present, C: Inferred Absent, D: Absent
Reasoning and evidence:

Liverani says the so-called "urban revolution" of the Uruk phase occurred 3800-3000 BCE.
Answer:
D
```

**Prompt part 7** Few-Shot CoT Examples For Institutional Variables

```
Question:
The characteristic 'Local-level administrators' is categorized under 'Local-level officials (provincial,
regional, civic administration)'. Was it present, inferred present, inferred absent, or absent for the
polity called 'Late Shang', during the time frame from 1250 BCE to 1045 BCE?
Options:
A: Present, B: Inferred Present, C: Inferred Absent, D: Absent
Reasoning and evidence:

Local rulers and elite families (i.e. the 'Archer Lords' and 'Princes'), who were only loosely
politically connected / answerable to the Shang kings, played a quasi-administrative role in ensuring
peace and maintaining productivity over the lands under their control.
Answer:
A

Question:
The characteristic 'Professionalization of local bureaucracy' is categorized under 'Local-level
officials (provincial, regional, civic administration)'. Was it present, inferred present, inferred
absent, or absent for the polity called 'Egypt - Dynasty I', during the time frame from 3100 BCE to 2900
BCE?
Options:
A: Present, B: Inferred Present, C: Inferred Absent, D: Absent
Reasoning and evidence:

The "precusor" central bureaucracy developed from Dynasty I
Answer:
B

Question:
The characteristic 'Executive power is separate or independent from judiciary' is categorized under
'Legal (formal) limits'. Was it present, inferred present, inferred absent, or absent for the polity
called 'Tocharians', during the time frame from 129 BCE to 29 CE?
Options:
A: Present, B: Inferred Present, C: Inferred Absent, D: Absent
Reasoning and evidence:

In Barfield's model of political power in the nomadic states of Central Asia, 'resolving disputes that
threatened internal order' is one of the roles of the leader, perhaps indicating a judicial function.

Answer:
C

Question:
The characteristic 'Local-level administrators' is categorized under 'Local-level officials (provincial,
regional, civic administration)'. Was it present, inferred present, inferred absent, or absent for the
polity called 'Jenne-jeno I', during the time frame from 250 BCE to 49 CE?
Options:
A: Present, B: Inferred Present, C: Inferred Absent, D: Absent
Reasoning and evidence:

Jenne-Jeno was a market and crafts center and before that perhaps an incipient market settlement. The
people were apparently without literacy and the archaeological record does not support the existence of
political or temple institutions. There is no evidence for an army that projected the power of this
polity into the surrounding regions. If some kind of executive did exist it likely would have been
without full-time administrators, and they would have been concerned only with the city of Jenne-Jeno.At
Jenne-jeno no evidence of "social ranking or authoritarian institutions such as a 'temple elite' has
been found.
Answer:
D
```

**Prompt part 8** Few-Shot CoT Examples For Social Mobility

```
Question:
The characteristic 'elite status is hereditary' is categorized under 'Status'. Was it present, inferred
present, inferred absent, or absent for the polity called 'Parthian Empire II', during the time frame
from 41 CE to 226 CE?
Options:
A: Present, B: Inferred Present, C: Inferred Absent, D: Absent
Reasoning and evidence:

Elite families such as the Surens and Karens had the greatest influence and probably held top posts such
as "satrap of satraps" and were regular satraps.
Answer:
A

Question:
The characteristic 'occupational mobility' is categorized under 'Status'. Was it present, inferred
present, inferred absent, or absent for the polity called 'British Empire II', during the time frame
from 1850 CE to 1968 CE?
Options:
A: Present, B: Inferred Present, C: Inferred Absent, D: Absent
Reasoning and evidence:

'social mobility' and 'occupational mobility' are closely-intertwined in Victorian society, so this is
my contribution to both categories. Primarily, mobility in this period was 'inter-generational,' and
social changes generally corresponded to professional changes in various ways. Movement up or down the
social hierarchy usually would only be small within a single lifetime, but steady inter-generational
shifts were not uncommon throughout this period, in every class.
Answer:
B

Question:
The characteristic 'elite status is hereditary' is categorized under 'Status'. Was it present, inferred
present, inferred absent, or absent for the polity called 'Cahokia - Late Woodland II', during the time
frame from 450 CE to 600 CE?
Options:
A: Present, B: Inferred Present, C: Inferred Absent, D: Absent
Reasoning and evidence:

No evidence for an increase in social complexity and hierarchy or deviation from the "trend toward
household autonomy" at this time.
Answer:
C

Question:
The characteristic 'Chattel slavery' is categorized under ' Proportion of population enserfed '. Was it
present, inferred present, inferred absent, or absent for the polity called 'Kingdom of Hawaii -
Kamehameha Period', during the time frame from 1778 CE to 1819 CE?
Options:
A: Present, B: Inferred Present, C: Inferred Absent, D: Absent
Reasoning and evidence:

In late precontact and early contact-era Hawai'i: 'The papa kauwā, at the bottom of the social scale,
are sometimes translated in Western literature as "slaves," but a better term is probably "outcast."'.
Answer:
D
```

**Prompt part 9** Few-Shot CoT Examples For Warfare

```
Question:
The characteristic 'Shields' is categorized under 'Armor'. Was it present, inferred present, inferred
absent, or absent for the polity called 'Latium - Bronze Age', during the time frame from 1800 BCE to
900 BCE?
Options:
A: Present, B: Inferred Present, C: Inferred Absent, D: Absent
Reasoning and evidence:

Weapons, statuettes, and "double shields" found in male burials suspected to infer elite military or
religious status.
Answer:
A

Question:
The characteristic 'Battle axes' is categorized under 'Handheld weapons'. Was it present, inferred
present, inferred absent, or absent for the polity called 'Magadha', during the time frame from 450 CE
to 605 CE?
Options:
A: Present, B: Inferred Present, C: Inferred Absent, D: Absent
Reasoning and evidence:

Reference for northern India in the 7th century CE: According to Hiuen Tsang (quoted here) some of the
Harsha infantry had 'Battle axes' and had been 'drilled in them for generations.'
Answer:
B

Question:
The characteristic 'Atlatl' is categorized under 'Projectiles'. Was it present, inferred present,
inferred absent, or absent for the polity called 'Yemen - Late Bronze Age', during the time frame from
1200 BCE to 801 BCE?
Options:
A: Present, B: Inferred Present, C: Inferred Absent, D: Absent
Reasoning and evidence:

These do not appear to be included in depictions of "warriors" in North Yemeni rock-art from the Late
Neolithic and Bronze Age, as reproduced in Jung (1991).
Answer:
C

Question:
The characteristic 'Dogs' is categorized under 'Animals used in warfare'. Was it present, inferred
present, inferred absent, or absent for the polity called 'Archaic Basin of Mexico', during the time
frame from 6000 BCE to 2001 BCE?
Options:
A: Present, B: Inferred Present, C: Inferred Absent, D: Absent
Reasoning and evidence:

Hassig lists war dogs among the new military "technologies" the Spanish introduced to the region in the
sixteenth century
Answer:
D
```

**Prompt part 10** Few-Shot CoT Examples For Cults and Rituals

```
Question:
During the time frame from 1568 CE to 1603 CE, was the characteristic 'Mutilation', associated with the
cults and rituals held by people of the 'Japan - Azuchi-Momoyama' polity, present, inferred present,
inferred absent, or absent?
Options:
A: Present, B: Inferred Present, C: Inferred Absent, D: Absent
Reasoning and evidence:

The high casualty rate of the Joseon and Ming forces, and the large number of ears collected during the
campaign was enough to build a large mound near Hideyoshi's Great Buddha, called the Mimizuka ("Mound of
Ears").
Answer:
A

Question:
During the time frame from 750 CE to 900 CE, was the characteristic 'Orthodoxy checks', associated with
the cults and rituals held by people of the 'Cahokia - Emergent Mississippian I' polity, present,
inferred present, inferred absent, or absent?
Options:
A: Present, B: Inferred Present, C: Inferred Absent, D: Absent
Reasoning and evidence:

Ethnohistoric accounts of similar rituals mention the presence of "war priests", as well as of war
leaders praising warriors for their "observance of purity"
Answer:
B

Question:
During the time frame from 1603 CE to 1868 CE, was the characteristic 'fasting', associated with the
cults and rituals held by people of the 'Tokugawa Shogunate' polity, present, inferred present, inferred
absent, or absent?
Options:
A: Present, B: Inferred Present, C: Inferred Absent, D: Absent
Reasoning and evidence:

don't mention fasting, then it almost certainly wasn't part of the ritual. Indeed, in Picken's
Historical Dictionary of Shinto, "fasting" is only mentioned once, in reference to the practices of
mountain ascetics
Answer:
C

Question:
During the time frame from 2270 BCE to 2083 BCE, was the characteristic 'risk of death', associated with
the cults and rituals held by people of the 'Akkadian Empire' polity, present, inferred present,
inferred absent, or absent?
Options:
A: Present, B: Inferred Present, C: Inferred Absent, D: Absent
Reasoning and evidence:

Not mentioned by "detailed" source
Answer:
D
```

**Prompt part 11** Few-Shot CoT Examples For Equity

```
Question:
The characteristic 'Property can be owned by people from all social classes' is categorized under
'Discrimination'. Was it present, inferred present, inferred absent, or absent for the polity called
'Hawaii III', during the time frame from 1580 CE to 1778 CE?
Options:
A: Present, B: Inferred Present, C: Inferred Absent, D: Absent
Reasoning and evidence:

'Land ownership' was a complex issue in Hawaii of this time, but all levels of the hierarchy from the
commoners working the land to the ali'i nui at the top were considered to have some form of rights to
the land. The Board of Commissioners to Quiet Land Titles, appointed by Kamehameha III in 1846 as part
of the Māhele land reforms, conducted a 'careful examination of the traditional system of land rights'.
Answer:
A

Question:
The characteristic 'Bureaucratic positions are open to both males and females' is categorized under
'Discrimination'. Was it present, inferred present, inferred absent, or absent for the polity called
'Elam - Crisis Period', during the time frame from 1100 BCE to 900 BCE?
Options:
A: Present, B: Inferred Present, C: Inferred Absent, D: Absent
Reasoning and evidence:

An earlier seal excavated at Susa and dated to the 'Middle or Neo-Elamite period, late 2nd-early 1st
millennium B.C.' also depicts a female audience scene, implying some continuity in administrative
practices through the first half of the 1st millennium BCE.
Answer:
B

Question:
The characteristic 'Bureaucratic positions are open to both males and females' is categorized under
'Discrimination'. Was it present, inferred present, inferred absent, or absent for the polity called
'Egypt - Dynasty II', during the time frame from 2900 BCE to 2687 BCE?
Options:
A: Present, B: Inferred Present, C: Inferred Absent, D: Absent
Reasoning and evidence:

There was a "precusor" bureaucracy in this period
Answer:
C

Question:
The characteristic 'Promotion in administrative positions is open to both males and females' is
categorized under 'Discrimination'. Was it present, inferred present, inferred absent, or absent for the
polity called 'Canaan', during the time frame from 2000 BCE to 1175 BCE?
Options:
A: Present, B: Inferred Present, C: Inferred Absent, D: Absent
Reasoning and evidence:

We know of only a single example in the thousand-year period of a Canaanite woman in a position of
independent power-a city ruler who sent two of the Amarna Letters, who called herself the "Mistress of
the Lionesses." We have no other details beyond that, and the location of her city is unclear, though a
find in 2009 led some scholars to speculate that she might have ruled Beit Shemesh.
Answer:
D
```

**Prompt part 12** Few-Shot CoT Examples For Economy variables (polity-level)

---

```
Question:
The characteristic 'Tribute' is categorized under 'State Income'. Was it present, inferred present,
inferred absent, or absent for the polity called 'Ptolemaic Kingdom I', during the time frame from 305
BCE to 217 BCE?
Options:
A: Present, B: Inferred Present, C: Inferred Absent, D: Absent
Reasoning and evidence:

possessions outside of Egypt (Syria, Cyprus, Cyrenaica, and parts of Asia Minor and Greece) were great
source of revenue for Ptolemies while in the state's possession. Perhaps not really 'tribute' as much as
colonial taxation, but was certainly a key economic resource  for the Ptolemaic rulers in addition to
taxation of surplus production from Egypt proper. Importantly, Ptolemaic control over these territories
were periodically disrupted during this period (during the Syrian wars, e.g.)
Answer:
A

Question:
The characteristic 'Polity has obligation to other polity or group' is categorized under ' Tribute '.
Was it present, inferred present, inferred absent, or absent for the polity called 'Roman Empire -
Dominate', during the time frame from 285 CE to 394 CE?
Options:
A: Present, B: Inferred Present, C: Inferred Absent, D: Absent
Reasoning and evidence:

payment to mercenary troops from 'barbarian' groups could be seen as attempt to pacify these groups and
give them steady source of revenue to stave off need for raiding
Answer:
B

Question:
The characteristic 'Property' is categorized under ' Degree of taxation '. Was it present, inferred
present, inferred absent, or absent for the polity called 'Cahokia - Sand Prairie', during the time
frame from 1275 CE to 1400 CE?
Options:
A: Present, B: Inferred Present, C: Inferred Absent, D: Absent
Reasoning and evidence:

It seems likely that there was no system of taxation during this period. There is 'little evidence of
elite activity' at Cahokia, and, more generally, most scholars believe there was population dispersal
out of the American Bottom area.
Answer:
C

Question:
The characteristic 'local elites' is categorized under ' Degree of taxation '. Was it present, inferred
present, inferred absent, or absent for the polity called 'Jenne-jeno IV', during the time frame from
900 CE to 1300 CE?
Options:
A: Present, B: Inferred Present, C: Inferred Absent, D: Absent
Reasoning and evidence:

, and for military defence. Djenne was a new city 2.5km south-east of Jenne-jeno - what relationship was
there - if any - between the old and the new cities? Diop (1987) says a "Sana-faran was their general-in-
chief"
Answer:
D
```

**Prompt part 13** Few-Shot CoT Examples For Well-Being

```
Question:
The characteristic 'Public markets' is categorized under 'Economic Well-Being'. Was it present, inferred
present, inferred absent, or absent for the polity called 'Medang Kingdom', during the time frame from
732 CE to 1019 CE?
Options:
A: Present, B: Inferred Present, C: Inferred Absent, D: Absent
Reasoning and evidence:

Already by the 8th and 9th centuries in central Java, there was a 'hierarchical marketing network' based
at the simplest level on periodic village markets known as pken or pkan.
Answer:
A

Question:
The characteristic 'Alimentary supplementation' is categorized under 'Public Goods'. Was it present,
inferred present, inferred absent, or absent for the polity called 'Egypt - Middle Kingdom', during the
time frame from 2016 BCE to 1700 BCE?
Options:
A: Present, B: Inferred Present, C: Inferred Absent, D: Absent
Reasoning and evidence:

Declaration of virtues for Intef "herald and governor under Thutmose III (Urk IV, 964-975)" (that also
reflects Middle Kingdom) includes: servant of the needy; father of the poor; guide of the orphan; mother
of the timid; shelter for the battered; guardian of the sick; husband of the widow; refuge for the
orphan.
Answer:
B

Question:
The characteristic 'Debt-relief measures' is categorized under 'Economic Well-Being'. Was it present,
inferred present, inferred absent, or absent for the polity called 'Jenne-jeno II', during the time
frame from 50 CE to 399 CE?
Options:
A: Present, B: Inferred Present, C: Inferred Absent, D: Absent
Reasoning and evidence:

At Jenne-jeno no evidence of "social ranking or authoritarian institutions such as a 'temple elite' has
been found.
Answer:
C

Question:
The characteristic 'Public markets' is categorized under 'Economic Well-Being'. Was it present, inferred
present, inferred absent, or absent for the polity called 'Hawaii I', during the time frame from 1000 CE
to 1200 CE?
Options:
A: Present, B: Inferred Present, C: Inferred Absent, D: Absent
Reasoning and evidence:

Hommon has argued that pre-contact Hawaiian society was marked by 'the absence or minimal development'
of various features, including markets.
Answer:
D
```

**Prompt part 14** Few-Shot CoT Examples For Legal System

```
Question:
The characteristic 'Multiple legal systems' is categorized under 'Procedures of Legal System'. Was it
present, inferred present, inferred absent, or absent for the polity called 'Ptolemaic Kingdom II',
during the time frame from 217 BCE to 30 BCE?
Options:
A: Present, B: Inferred Present, C: Inferred Absent, D: Absent
Reasoning and evidence:

Ptolemaic / Greek system coexisted with traditional 'Egyptian' system - mainly based on the language
used - and a separate court system for mixed cases involving both Greeks and Egyptians
Answer:
A

Question:
The characteristic 'Lease' is categorized under ' Private property (land) '. Was it present, inferred
present, inferred absent, or absent for the polity called 'Funan II', during the time frame from 540 CE
to 640 CE?
Options:
A: Present, B: Inferred Present, C: Inferred Absent, D: Absent
Reasoning and evidence:

In an inscription there is mention to the donation of land to a temple, but the conditions seem to imply
that the owner retained some kind of right over the land and that only the product was given to the
temple: "the land is reserved: the produce is given to the god".
Answer:
B

Question:
The characteristic 'Non-state expropriation' is categorized under 'Inheritance System'. Was it present,
inferred present, inferred absent, or absent for the polity called 'Medang Kingdom', during the time
frame from 732 CE to 1019 CE?
Options:
A: Present, B: Inferred Present, C: Inferred Absent, D: Absent
Reasoning and evidence:

Wisseman Christie's discussion of sima transfers implies that religious foundations as well as Javanese
rulers and local lords (rakai) respected villagers' rights to land, and would not take possession of it
without compensating them first: 'If actual title to land was to be transferred to a religious
foundation, then that land was first purchased, for gold, from the villagers affected'.
Answer:
C

Question:
The characteristic 'State expropriation' is categorized under 'Inheritance System'. Was it present,
inferred present, inferred absent, or absent for the polity called 'Kediri Kingdom', during the time
frame from 1049 CE to 1222 CE?
Options:
A: Present, B: Inferred Present, C: Inferred Absent, D: Absent
Reasoning and evidence:

Rights to communal wanua land were passed down within village communities, derived from 'deified
ancestors', and the state seems to have respected this: 'The wanua in this sense was acknowledged by the
court to have had clearly identified boundaries and to have included all of the land within those
borders, cultivated or not'.
Answer:
D
```

