# OpenReview forum: "Large Language Models' Expert-level Global History Knowledge Benchmark (HiST-LLM)"
_NeurIPS.cc/2024/Datasets_and_Benchmarks_Track — NeurIPS 2024 Track Datasets and Benchmarks Poster_

### Official Review · Reviewer_SzfT · 2024-07-23
**Solid & enlightening work - though an ethics review & discussion is needed**

**Rating:** 9
**Confidence:** 4
**Clarity:** The paper is very well written.

**Review:**

This is an enlightening work with solid evaluation protocols, a rigid data collection process and interesting results. I recommend publication at the venue and I believe it would be one of the best works in this year's cohort. There are some minor limitations on the breadth of the evaluation and details in the experiments, but they do not diminish the significance of the work. However, I also believe that it is important to have a discussion on ethical concerns, especially regarding inherent bias on English-language sources and Seshat's relationship with the study of cliodynamics.

**Strengths:**

* Studying the historical accuracy and awareness of language models is incredibly important and under-explored, considering the increasingly important role of which in our society. The authors appear to be well-positioned to study this topic, and the analysis in the work is well-backed.
* Detailed explanation on data collection
* Evaluation appears to be thorough, and the analysis is well grounded and insightful

**Additional Feedback:**

I'd love to see a wider discussion on historical knowledge of language models, even multi-modal language models, at this venue, hence I strongly recommend acceptance.

**Correctness:**

The claims appear to be correct and the experiment appear to be designed appropriately and performed correctly.

**Documentation:**

The dataset is properly documented.

**Ethics:**

There's considerable concern, according to the guidelines, in the following aspects as they relate to this work:

1. Security. Various authors of the work are affiliated with the study of cliodynamics, which is part of a much wider discussion with much debate [1,2,3]. The Seshat dataset appears to be very much based on a quantitative approach, and hence uninformed usage of which in critical benchmarks, datasets and models could pose very serious risk to misinterpreting events in the past and mispredicting events in the future.

2. Discrimination. Collecting data and constructing the Seshat dataset is done on English-language sources, which may greatly affect the objectiveness of various evaluations from influences like colonialism.

This is not to suggest that the work should not be published, or that the authors have not discussed these aspects in the work (particularly section 3.3). Nevertheless, I believe that a conversation & ethics review is needed, as a different stakeholder other than the Reviewers should provide perspectives on this work.

[1]: https://foreignpolicy.com/2021/06/20/history-cliodynamics-weird-turchin/
[2]: https://www.theguardian.com/books/2023/may/28/end-times-by-peter-turchin-review-elites-counter-elites-and-path-of-political-disintegration-can-we-identify-cyclical-trends-in-narrative-of-human-hope-and-failure
[3]: https://www.academia.edu/11832274

**Limitations:**

* 4-way multiple-choice eval is arguably a very naive and preliminary way of representing historical knowledge. I'm curious about LLM's capability in TriviaQA-style evaluations, either with or without context.
* The work touches briefly on using historical text corpora to pretrain or finetune LLMs (L 117-119) and existence of historical questions in popular evals (L 121-133). I suggest, if possible, that the authors do a review on whether existing training corpora contains historical text (e.g. RedPajama, Internet Archive dumps), and discuss the prospect of potential data pollution, or even check for it.

**Opportunities For Improvement:**

* What would be the performance of Anthropic's Claude 3 class of models, Google's Gemini models, and Mistral's models? The code artifacts appear to reference mistralai/Mixtral-8x7B-Instruct-v0.1 but the evaluation results are not mentioned in the paper.
* What would be the performance of LLMs using chain-of-thought prompting techniques? Would that result in statistically significant difference in performance?

**Relation To Prior Work:**

The paper has clearly discussed its relation to prior works.

**Summary And Contributions:**

The work curates a structured representation of human historical knowledge consisting of 36k data points, across all major historical societies and major periods of time. The dataset is constructed by the Seshat Board, a group of experts and scholars in historical science, some affiliated with the study of cliodynamics. The work then evaluates some commercially available language models in a 4-way multiple-choice evaluation scheme with multi-shot examples. The results indicate that leading LLMs generally consistently perform better than random guess but worse than experts, with discrepancies in accuracy in both periods of time and regions.

---

> ### Author Rebuttal · Authors · 2024-08-17
>
> Thank you for the positive feedback and suggestions. This strongly motivated and helped us to improve the manuscript this week and we are committed to improving it further. We reply to the  feedback below:
> # Additional models and chain-of-thought prompting
> We have revised our benchmark using chain-of-thought prompting and added custom few-shot prompts per question category. We also added three additional models to the benchmark (Please see table 1-2 and global rebuttal for details). Indeed, we originally wanted to include Mistral in the benchmark, but the endpoints for these models were unstable and slow. Currently we are using the provider together.ai, but are working on switching to include these models.
> # Reliance on balanced accuracy and 4-way multiple-choice does not test LLM’s reasoning.
> We agree with the reviewer that testing the LLMs performance with more diverse metrics would be an important way forward. While a full analysis of the reasoning outputs of chain-of-thought was not feasible within the revision period, we have saved these outputs and plan to analyze them further. Moreover, we also believe that the Seshat dataset would allow creating more complex tasks, including a wider range, open-ended questions that simulate natural conversations or indeed reading comprehension focused tasks like presented in TriviaQA.  Additionally analyzing a sample of the chain-of-thought reasonings of models could also provide useful insights into the reasoning mistakes models make. Any significant insights will be reported in future work to enhance the assessment of LLM performance.
> # Review of data contamination
> We thank the reviewer for this suggestion. We agree it would be very interesting to check data pollution. However, to our knowledge, training corpora of the LLMs in this benchmark are not generally available, complicating this exercise. Moreover, because currently our benchmark is primarily about facts and not reasoning, and thus we believe training data “contamination” or “pollution” is less of a concern: a correct answer to any question implies a representation of a specific historic fact in the LLM, presumably coming from its training data. However, we agree that better characterizing how facts encountered during training gets represented and retrieved in LLMs is an important question and our dataset can help future research in this direction, by (1) presenting a large amount of facts representing specialized knowledge and thus likely only present in any generic corpus with low frequency; and (2) providing citations to specific sources of facts that could be further evaluated based on their inclusion or exclusion of further studies that perform model training. We will add these points in section 6.
> # Cliodynamics and Quantitative approaches
> We thank the reviewer for communicating this concern. We were not able to view the references, but if you could please provide them we would be happy to include them in the manuscript and discuss them further.  We acknowledge that there have been scholarly debates on the applicability and use of quantitative methods in history. These have been very fruitful in delineating the possibilities and limitations, along with fostering the exchange of ideas and collaborations among researchers with very different backgrounds. E.g. for a review of issues and solutions for creating systematic datasets for quantitative analysis, see [1]. In the current work, we excluded all data points marked as scholarly disagreement, carefully evaluated possible issues, and believe the Seshat dataset is well suited to be used as factual knowledge on the specific topics and questions that are included in it in a benchmark setting. We also clarify that while Seshat machine-readable codes are well suited for cliodynamic analysis, the dataset use is not exclusive for this purpose and can be used in other quantitative and qualitative studies (for example, qualitative studies may find the collection of references and justifications useful). We agree that we should clarify the distinction between cliodynamics and Seshat. We also agree that  uninformed use of the dataset could lead to problematic results and will update the manuscript as outlined in the global rebuttal. We thank you for highlighting these concerts and reiterate are very open to extend this part as well as incorporate additional references in the manuscript.
>
>
> “While Seshat takes special care in the data collection process, providing detailed qualitative justifications with each data point overseen by a Data Review Board, the dataset may still simplify important historical knowledge (see [1] Slingerland et al. 2020;  for issues and solutions in creating systematic datasets for cultural comparison). Therefore, using this dataset for historical analysis and interpretation should only be done in collaboration with relevant experts. While the Seshat dataset has been used in several quantitative Cliodynamic \footnote{ Cliodynamics is an interdisciplinary field that combines historical analysis and mathematical modelling. For more details and criticism see [2,3,4,5] }
>  it remains an independent dataset that can be used for different research approaches and is not tied to any single theoretical framework.”
> References:
> 1. Slingerland et al. 2020 doi.org/10.1017/ehs.2020.30.
> 2. Beheim et al. 2019 doi.org/10.1038/s41586-019-1043-4
> 3. Whitehouse et al. 2022 doi.org/10.1080/2153599X.2022.2074085
> 4. Hoyer et al. 2023, doi.org/10.1098/rstb.2022.0402
> 5. Jaeweon et al. 2020, doi.org/10.1038/s41467-020-16035-9
> # English language and discrimination
> We agree that this is one of the strongest limitations of our work. We will modify the draft to discuss these issues as further outlined in the global rebuttal and are very open to taking further suggestions.
> # Additional feedback
> Thank you very much for the positive feedback. We agree this work is just a start of the conversation and we look forward to expanding the work further.

---

### Official Review · Reviewer_YNxR · 2024-07-24
**Large historical database**

**Rating:** 6
**Confidence:** 3
**Correctness:** Benchmarking could be improved, see l…
**Clarity:** Yes

**Review:**

# Summary
In this manuscript, the authors introduce a dataset of human history across thousands of years, and hundreds of societies. The manuscript also provides a benchmark of several LLMs on said dataset.

# Strong points
1. The dataset is rather large, and data has been collected through a properly reviewed process
2. Analysis of the results is varied, and presents several interesting points, e.g., historical periods of model inaccuracy

# Weak points
1. Accuracies are rather low, but I suspect that this may also come from the prompting, which perhaps should have been studied more in depth
2. There's no study of the relationship w/ training data: do models performing better on one, rather than another, historical period have had more access to it? Same, e.g., for regional performance

# Recommendation
Weak accept.

**Strengths:**

# Strong points
1. The dataset is rather large, and data has been collected through a properly reviewed process
2. Analysis of the results is varied, and presents several interesting points, e.g., historical periods of model inaccuracy

**Additional Feedback:**

.

**Documentation:**

Yes

**Limitations:**

Yes

**Opportunities For Improvement:**

# Weak points
1. Accuracies are rather low, but I suspect that this may also come from the prompting, which perhaps should have been studied more in depth
2. There's no study of the relationship w/ training data: do models performing better on one, rather than another, historical period have had more access to it? Same, e.g., for regional performance

**Relation To Prior Work:**

Yes

**Summary And Contributions:**

In this manuscript, the authors introduce a database of human history across thousands of years, and hundreds of societies. The manuscript also provides a benchmark of several LLMs on said dataset.

---

> ### Author Rebuttal · Authors · 2024-08-17
>
> We thank the reviewer for the critical feedback and outline how we incorporate it to improve the manuscript.
> # Low accuracy may be due to prompting
> We thank the reviewer for pointing this out. While we consider the low accuracy a result indicating the LLMs current limitations with respect to historical knowledge rather than an issue, we agree that the lower accuracies may be influenced by the prompting strategy used and warrants a further evaluation.
> To explore this further, we conducted additional experiments with different prompting strategies. Specifically,  we expanded content framing of prompts by using 4 tailored examples for each variable category. We also switched to chain-of-thought prompting, asking models to provide reasoning before their answers. Results are mixed, while GPT-4-turbo and 4o increased their performance by 2.2% and 2.7% respectively, GPT-3.5 and Llama-3-70B’s performance decreased by 1.9% and 1.8% respectively. All changes are statistically significant (see Tables 1 and 2 and Figure 1 of the supplementary PDF for the chair-of-thought results).  These results suggest that while prompting strategies do have an effect, they do not significantly alter the overall limitations of the models as revealed by our benchmarks. We will include these findings in the revised manuscript. We are also working on other prompt engineering techniques and will update with robustness checks if allowed to revise.
>
> # Relationship between performance in regions and training data
> This is a very interesting and relevant question, as we expect that training data will affect models’ performance to a large degree. While we can speculate that the specialized knowledge included in Seshat is not present in training corpora due to models’ low performance, it would be interesting to explore the extent of overlap. Unfortunately,  exploring this relationship in depth is challenging due to the limited transparency in the data and methodologies used for training most LLMs. A plausible way forward would be to look into availability of data in openly available datasets. For example, one could count occurrences of polities or regions in Wikipedia text dumps. We are currently looking into this and would report any findings or at the least suggest further work. For example, by adding in the discussion:
>
> "Moreover, future work could explore the relationship between LLM training data and their performance on specialized datasets like Seshat, to better understand the influence of training data on model accuracy."

---

### Official Review · Reviewer_oaF3 · 2024-07-25
**A valuable benchmark for assessing LLMs' historical knowledge using the comprehensive Seshat Global History Databank**

**Rating:** 6
**Confidence:** 3
**Correctness:** Yes
**Clarity:** Yes

**Review:**

Pros:
- Provides a robust benchmark for evaluating LLMs' historical knowledge, filling a gap in existing research.
- Offers a valuable resource for historians, archaeologists, and researchers to assess and improve LLMs' performance in processing historical data. Promotes interdisciplinary applications, enhancing the potential impact of LLMs in humanities and social sciences.
- Employs a rigorous and well-documented methodology, ensuring reliability and validity of findings. Detailed analysis of LLMs' performance across different regions and time periods provides valuable insights.
- Relies on data assembled and reviewed by history experts, enhancing the credibility and authority of the benchmark.

Cons:
- Evaluates only a limited number of LLMs, potentially missing insights from other prominent models.
- The dataset primarily uses English-language sources, which may result in less comprehensive coverage for non-English-speaking regions and introduce bias.
- The paper does not deeply explore practical applications or implications of the benchmark findings for real-world use cases.
- Relies heavily on balanced accuracy and multiple-choice questions, which might not fully capture the nuances of LLMs' understanding and reasoning abilities. More diverse metrics and evaluation methods could provide a more comprehensive assessment.

**Strengths:**

- By using the Seshat Global History Databank, the study provides a valuable resource that can be utilized by historians, archaeologists, and researchers in related fields to assess and enhance LLMs' capabilities in processing historical data.
- The research methodology is rigorous and well-documented. The use of a carefully curated dataset and the systematic approach to evaluating LLMs' performance ensures the reliability and validity of the findings.
- The study introduces a robust and comprehensive benchmark for evaluating LLMs' historical knowledge, addressing a gap in current research.

**Additional Feedback:**

None

**Documentation:**

Yes

**Limitations:**

- While the study focuses on GPT-3.5, GPT-4-Turbo, GPT-4o, and LLama-3-70b, it does not evaluate other prominent LLMs that could offer additional insights into the comparative performance across a broader range of models.
- The Seshat Databank, while comprehensive, primarily uses English-language sources. This may result in less comprehensive coverage for non-English-speaking regions and potentially introduce bias in the dataset.
- The study's focus on historical knowledge may limit its immediate applicability to other fields. The findings might not directly translate to improvements in LLM performance in other domains, such as contemporary events or scientific knowledge.

**Opportunities For Improvement:**

- Include more non-English language sources in the Seshat Databank to ensure a more comprehensive and balanced representation of global historical knowledge.
- Expand the evaluation to include other prominent LLMs beyond GPT-3.5, GPT-4-Turbo, GPT-4o, and LLama-3-70b. This would provide a more comprehensive comparison and potentially uncover additional insights.
- Simplify the descriptions of the data conversion and evaluation processes to make the paper more accessible to a broader audience, including those not familiar with specific methodologies used.

**Relation To Prior Work:**

Yes

**Summary And Contributions:**

This paper introduces a benchmark for assessing the historical knowledge of Large Language Models (LLMs) using the Seshat Global History Databank, which contains structured data from over 600 societies across major world regions and historical periods. The study evaluates the performance of GPT-3.5, GPT-4-Turbo, GPT-4o, and LLama-3-70b through multiple-choice questions derived from the dataset. Results indicate that LLMs perform better than random guessing but fall short of expert-level comprehension, with performance varying by region and time period. The findings highlight potential biases in LLM training data and suggest that the Seshat dataset can help improve LLMs for historical research.

---

> ### Author Rebuttal · Authors · 2024-08-17
>
> We thank the reviewer for the thoughtful and constructive feedback. Below, we address each of the points raised and how we incorporate the suggestions.
> # Evaluation of Additional LLMs
> We agree that expanding the evaluation to include more LLMs would provide a more comprehensive comparison and potentially uncover additional insights. In response to this suggestion, we have added Gemini-1.5-flash and Llama-3.1 8B(FP8) and 70B(FP8) to the benchmark. We are working towards and have applied for internal university funding for a computer cluster to add LLaMA 3.1 450B, Gemini-1.5-pro, and Mixtral models. The results of these evaluations will be included in the final paper (see Table 1,2 and Figure 1 of the supplementary PDF).
> # Discuss practical applications and implications.
> We thank the reviewer for suggesting we add a discussion on the applications and implications of the benchmark.  We will add the following to Section 6:
>
> “As LLMs are increasingly used in information retrieval and search solutions, being aware of and monitoring limitations and biases of embedded knowledge is of crucial importance. Our work enables a range of such applications by providing a thorough ground truth dataset that is especially amenable for further automated processing and incorporation in ML applications.”
> # Reliance on balanced accuracy and multiple-choice does not assess reasoning.
> We agree that balanced accuracy and multiple-choice questions may not fully capture the nuances of LLMs' reasoning abilities. To understand this better, we conducted additional benchmarks using chain-of-thought prompting, which improved performance slightly (~2%) but did not significantly alter our findings. These results are included in the revised tables. While a full analysis of the reasoning outputs of chain-of-thought was not feasible within the revision period, we have saved these outputs and plan to analyze them further. Any significant insights will be reported in future work to enhance the assessment of LLM performance.
> # English-Language Sources
> We acknowledge that the predominance of English-language sources in the Seshat dataset is a strong limitation. While the dataset has been compiled with the input of experts from diverse backgrounds, who often have access to non-English sources, we recognize that biases could remain, particularly if dissenting interpretations are only available in non-English literature. The Seshat team is currently working on expanding the number of non-English sources. However, given that creating the current dataset based mostly on English-language sources took over a decade of work and significant resources, we believe that reviewing Seshat data on a large scale based on additional (non-English) literature is infeasible for the near short term. To address this limitation, we will edit the text as outlined in the global rebuttal. In particular:
>
> Add to the second paragraph of section 3.2 Data collection
>
> “While the language of these scholarly publications varies, there is a predominance of English sources. This mostly is due to the inherent imbalance in the language of history scholarship, which is increasingly in English. The Seshat team is currently working on expanding the number of Non-English sources”
>
> Modify section 3.3 Data limitations and ethical considerations to add
>
> “First, the data was collected mostly from English-language sources. This limitation likely results in less comprehensive coverage for non-English-speaking regions and limits diversity which is an ethical consideration and area of improvement.”
>
> We will also quote Seshat’s Research Ethics statement
>
> “The Seshat project involves the study of myriad different communities and populations from the past. Some peoples living today trace their ancestry to one or more of these past groups. As researchers, we have an obligation to present fair-minded, responsible, and respectful information concerning the past. While maintaining a commitment to scientific enquiry, we are committed to avoiding biased interpretation or representation of past or contemporary cultures, to refraining from using harmful or disrespectful terminology, and to treating sensitive information or topics with appropriate nuance and respect for the dignity and lived experiences of descendant communities.”
>
> Modify section 6, Discussion and Conclusions
>
> Add this text after the second paragraph:
>
> “This work presents a first benchmark of LLMs expert-level history knowledge. Going forward this should be taken as a lower bound, particularly for regions and countries such as Latin America, China, Japan, Egypt, and the Middle East with a wider scholarship on languages other than English. The Seshat team continues to expand the data sources with explicit efforts to consult and cite non-English literature. Important future work includes increasing collaborations with universities in the global south and indigenous groups. These avenues of further work can ultimately provide a more stringent and higher bar benchmark that LLMs’ performance should be measured with.”
> # Simplify the description of the data conversion process.
> We thank the reviewer for pointing this out. We will revise section 4, as shown below and add an in depth discussion of our data conversion and evaluation to the appendix.
>
> “To benchmark the LLMs’ historical knowledge, we converted the dataset into multiple-choice questions and prompted LLMs to answer these. Specifically, we asked whether a certain variable (e.g., writing) was present, absent, inferred present, or inferred absent in a particular polity and time frame. We use a multi-shot approach, providing four examples demonstrating solutions to other questions, in order to aid the models understanding of the task and additionally ask the model to provide its own reasoning before giving an answer. The exact prompts are shown in Appendix B. We also used personification, instructing the LLMs to act as history experts to enhance their performance.”

---

### Official Review · Reviewer_5Gsk · 2024-08-02
**Well organized benchmark with unclear licensing and limited evaluation**

**Rating:** 7
**Confidence:** 2
**Clarity:** The paper is well-written.

**Review:**

The dataset proposed in this work is potentially useful for evaluating LLMs across different time periods and regions, although the fact that the data are mostly in English is a bit concerning from a bias perspective. It is unclear how this influences the evaluation, although I concede that this is potentially a larger problem beyond the scope of this benchmark. What is potentially more concerning, however, is the unclear licensing of the dataset -- the authors note this in section 3.3. Beyond these two issues, the evaluation seems limited to four LLMs and no ablation across scales. Aside from these potential issues, the paper is organized and written well, the evaluation procedure seems rigorous, and the dataset itself is organized in a useful manner, with region and time-period based categories with which researchers can ablate their results.

**Strengths:**

- The description of the collected dataset is thorough and presented in a helpful way.
- The evaluation procedure seems rigorous, and includes 95% confidence intervals over all results.
- The idea to segment the evaluation by region and by times is interesting and a solid contribution. It's helpful to see how LLMs have region-based and time-period-based failure modes.

**Additional Feedback:**

Table 2 (as well as 1 and 3) are difficult to parse -- it would be helpful for the authors to bold the results or simply present the results and confidence intervals as a figure as opposed to a raw numerical table.

**Correctness:**

The major technical claims appear to be correct, with the caveat that it is unclear how the fact that the sources are mostly in English impacts the conclusions.

**Documentation:**

Sufficient documentation is provided.

**Ethics:**

The proposed dataset primarily contains English-language sources, which conceivably could lead to some of the conclusions in the paper. Additionally, the copyright considerations in the work are unclear. Both of these points are noted in the paper itself, in section 3.3.

**Limitations:**

The authors discuss limitations in 3.3.

**Opportunities For Improvement:**

- It is unclear how non-LLMs perform on these tasks. For example, how do human experts perform? What about an LLM equipped with an external database to search for answers? How do solutions based on Google search perform? These are just examples, but I feel that the evaluation could be improved by including an additional baseline showing that LLMs as a whole fall short or that even though LLMs aren't particularly good at these tasks.
- More generally, the evaluation seems to only include four LLMs. The evaluation could be strengthened by including a larger variety of LLMs or by evaluating them at different scales.
- The fact that the dataset suffers from unclear copyright is a potential significant issue for this benchmark, per section 3.3.

**Relation To Prior Work:**

The authors adequately discuss related work, including the various ways in which the proposed benchmark differs from prior work.

**Summary And Contributions:**

This paper proposes a databank of global history facts used to evaluate LLMs. The information in the databank, and the analyses in the paper, is categorized by geographical region and primarily contains facts that go beyond common knowledge, and focuses more on graduate or expert-level knowledge of historical along with how this information was derived (via direct factual evidence or otherwise). The authors conclude that GPT and Llama models seem to do better on Latin America, Caribbean, and North American histories, underperform on facts related to Sub-Saharan Africa, and that there is overall less historical data for the global south.

---

> ### Author Rebuttal · Authors · 2024-08-17
>
> We thank the reviewer for the thorough assessment and suggestions for improvements. We believe we have outlined how we addressed all concerns and incorporated suggestions in the global rebuttal and give additional details in this section.
>
> # Larger variety of LLMs and scales.
>
> We thank the reviewer for this suggestion. We have repeated the analysis on three additional models: Gemini-1.5-flash, Llama 3.1 8B(FP8) and Llama 3.1 70B(FP8). The results of these evaluations will be included in the final paper (see Table 1 and Figure 1) of the supplementary PDF. Additionally we are working towards and have applied for internal university funding for a computer cluster to add LLaMA 3.1 450B (FP8), Gemini-1.5-pro, and Mistral models.
>
> # Copyright issues
>
> We thank the reviewer for pointing this out. In hindsight we realize that section 3.3 was unclear on whether the potential copyright issues were with the Seshat dataset or to future text data that could be collected using the Seshat dataset. We clarify that the Seshat dataset has no licensing issues, and these issues may only arise if users take text from non-open-source references in the dataset. This dataset follows the Seshat project guidelines and all data is published under a creative commons license, as described on the project downloads page. Therefore, we will remove the last sentence of section 3.3 or clarify as follows:
>
> “This dataset is released under creative commons license. Going beyond the current dataset and scraping text from cited references to train models should only be done in consultation with a legal team and ethical board approval to avoid infringing copyright of the cited references.”
>
> # Predominance of English source
> We thank the reviewer for pointing these out and agree it is a major limitation. As outlined in the general rebuttal, we will modify the draft (sections 3.2, 3.3, and 6) to underscore this issue, the limitations, and our current work to expand to non-english sources.
>
> # Non-LLMs’ performance. Humans, RAG, Google search.
>
> Measuring Human performance in expert level knowledge of global history would be very interesting, but poses the question which humans? If we focus on high school, undergraduate or master level, we would require a stratified international sample of participants (regions covered in a history curriculum are highly dependent on the country of education). We also anticipate low performance given that our dataset covers expert level knowledge. If we focus on expert level knowledge (PhD, Postdocs, and Professors), then this is precisely what Seshat aims to capture: it represents the combined knowledge of human experts (while certain questions include scholarly disagreement, as recorded in Seshat as well, these were excluded from our benchmarks). One could design a study to measure the performance of advanced PhD students or Postdocs. However, academic knowledge is often unevenly distributed and based on individual scholars’ interests, which can influence benchmarks involving human experts. Thus, any such evaluation would require a careful selection of participants. Finally, this study would require a considerable amount of resources since the test would be time consuming and we would follow fair payment practices and pay participants Europe-level hourly wages of PhD and Postdocs.
>
> Equipping LLMs with external databases is a very promising suggestion to improve performance. However, choosing the external database is not simple. As discussed in section 2.1, Seshat is unique in its global coverage and provision of references. Therefore, the only suitable database may be the Seshat Zotero library. Even after restricting the library to the references relevant for this benchmark, there would be over 600 articles and whole books. Feeding these texts to LLMs would require implementing state-of-the-art RAG techniques and extensive compute resources. Especially since information required to answer our questions would require understanding long range dependencies in book chapters and retrieving multiple sections from very long documents. Moreover, this experiment would not be a benchmark solely of expert-level knowledge, but include information retrieval abilities.
>
> The suggestion for comparison with Google search is also very interesting, given also the context that LLMs are expected by many to be replacing traditional web search solutions for finding information. However, defining precisely how to use Google search as a benchmark is difficult. If only as a tool, this is similar to how Seshat has been built, by having RAs go through google scholar searching and reading related articles (notably its taken over 10 years). A mix between Google search and LLMs may be more promising, but designing the exact structure and methodology would require a considerable amount of work. We are very open to suggestions for further work and outlining them in the draft too. Indeed, one of the main motivations behind this project was to understand how LLMs could be used to expand Seshat.
>
> Overall, we agree that all these benchmarks would be very interesting, but as explained above, doing these experiments with an acceptable level of quality would require a vast amount of work and resources that go beyond the scope of the present project.
>
> # Additional feedback
>
> Thank you for your suggestions on how to make our results clearer. We will incorporate this feedback as outlined in the tables and plots in the new pdf (best results in tables are now bolded) and we also illustrate the results with bar plots and error bars.

---

> > ### Comment · Reviewer_5Gsk · 2024-08-28
> >
> > Thank you for the additional results and for the clarifications surrounding the ethical issues that I mentioned! I am satisfied with the response and I will raise my score accordingly.

---

### Author Rebuttal · Authors · 2024-08-17

# Thanks
We thank all reviewers for their time and detailed feedback. We outline how we incorporate this feedback below to substantially increase the quality of the manuscript.

# Additional models
We have added Gemini-1.5-flash, Llama-3.1 8B(FP8) and Llama-3.1 70B(FP8) to the benchmark (see Table 1, 2 and Figure 1). We are working towards and have applied for internal university funding to add LLaMA 3.1 450B, Gemini-1.5-pro, and Mistral models.
# Prompt Engineering

We acknowledge the limited prompt engineering in the initial paper, which used basic contextual framing (e.g., role-playing as an 'expert in history' with 4-shot examples). We have expanded content framing of prompts by using 4 tailored examples for each variable category. We also switched to chain-of-thought prompting, asking models to provide reasoning before their answers. Results are mixed, while GPT-4-turbo and 4o increased their performance by 2.2% and 2.7% respectively, GPT-3.5 and Llama-3-70B’s performance decreased by 1.9% and 1.8% respectively. All changes are statistically significant.

# Licensing clarification
We clarify that the benchmark and dataset have no copyright or licensing issue. The potential for copyright infringement mentioned in Section 3.3 refers to a potential future scenario in which users take references included in our dataset, scrape a considerable amount of text, and use it to train LLMs. Since we believed this to be a research direction enabled by our dataset, and given recent lawsuits to OpenAI about training data use, we wanted to be cautious and flag the potential issue. In hindsight, we realize the wording was confusing and unnecessary given that it falls beyond our benchmark and dataset. Therefore, we will clarify or remove this part (last sentence in 3.3).

# English language predominance

We agree with the reviewers that Seshat having predominantly English Sources is one of the strongest limitations. We will emphasize this and how it may affect results adding to sections or modifying as follows

Section 3.2:

“While the language of these scholarly publications varies, there is a predominance of English sources. This mostly is due to the inherent imbalance in the language of history scholarship in the Global North, which is increasingly in English. The Seshat team is currently working on expanding the number of Non-English sources”

Section 3.3:

“First, the data was collected mostly, though not exclusively, from English-language sources. This limitation likely results in less comprehensive coverage for non-English-speaking regions and limits diversity which is an ethical consideration and an area of improvement.”

We will also add the Seshat’s Research Ethics statement:

“The Seshat project involves the study of myriad different communities and populations from the past. Some peoples living today trace their ancestry to one or more of these past groups. As researchers, we have an obligation to present fair-minded, responsible, and respectful information concerning the past. While maintaining a commitment to scientific enquiry, we are committed to avoiding biased interpretation or representation of past or contemporary cultures, to refraining from using harmful or disrespectful terminology, and to treating sensitive information or topics with appropriate nuance and respect for the dignity and lived experiences of descendant communities.”

Section 6:

“This work presents a first benchmark of LLMs expert-level history knowledge. Going forward this should be taken as a lower bound, particularly for regions and countries such as Latin America, China, Japan, Egypt, and the Middle East with a wider scholarship on languages other than English. The Seshat team continues to expand the data sources with explicit efforts to consult and cite non-English literature. Important future work includes increasing collaborations with universities in the global south and indigenous groups. These avenues of further work can ultimately provide a more stringent and higher bar benchmark that LLMs’ performance should be measured with.”

# Non-LLMs’ performance
We thank R5Gsk for suggesting to compare the current benchmark with the performance of humans, RAG-LLMs, and google search. There are excellent questions, but as we argue in more detail in the rebuttal 5Gsk, making these benchmarks with an acceptable level of quality would require substantial work and resources that lie outside the scope of the current paper. To outline this future direction we add to section 6.

“Finally, assessing global historical knowledge across diverse populations, with balanced representation of societies worldwide, would be a valuable study for the humanities and provide a meaningful benchmark for comparison with LLMs.”
# Geography
We thank ER LHD6 for rightly pointing out an inaccuracy in our description of regions. We will add a map with regions as suggested, see in Fig. 2 and the following clarification to section 3.

“We use the UN regions as a baseline for studying the performance across geography, but make two modifications. (i) We split North America and Europe into two regions; (ii) We include Hawaii in Oceania. We made these modifications since most of the focus of Seshat is on the polities that existed before colonization and Europe and North America have very different historical pathways in that period. Similarly, the first inhabitants of Hawaii are of Polynesian descent and thus until the nineteenth century, its history was more related to regions in Oceania.”

# Cliodynamics and Quantitative approaches
We thank the reviewers for communicating the concerns. We will add the requested discussion to the manuscript. For the precise text please see rebuttal to Reviewer SzfT.

---

### Decision · Program_Chairs · 2024-09-26

**Decision:**

Accept (Poster)

**Comment:**

The reviewers unanimously found the paper to be of high quality and significance for acceptance.